# Chromatin environment-dependent effects of DOT1L on gene expression in male germ cells
Manon Coulée[1], Alberto de la Iglesia [1], Mélina Blanco [1,2,3], Clara Gobé[1], Clémentine Lapoujade[2,3], Côme Ialy-Radio[1], Lucia Alvarez-Gonzalez[4,5], Guillaume Meurice[6], Aurora Ruiz-Herrera [4,5], Pierre Fouchet[2,3], Julie Cocquet [1,8] ✉ & Laïla El Khattabi [1,7,8] ✉

The H3K79 methyltransferase DOT1L is essential for multiple aspects of mammalian development where it has been shown to regulate gene expression. Here, by producing and integrating epigenomic and spike-in RNA-seq data, we decipher the molecular role of DOT1L during mouse spermatogenesis and show that it has opposite effects on gene expression depending on chromatin environment. On one hand, DOT1L represses autosomal genes that are devoid of H3K79me2 at their bodies and located in H3K27me3-rich/H3K27ac-poor environments. On the other hand, it activates the expression of genes enriched in H3K79me2 and located in H3K27me3-poor/H3K27ac-rich environments, predominantly X chromosome-linked genes, after meiosis I. This coincides with a significant increase in DOT1L expression at this stage and a genome-wide acquisition of H3K79me2, particularly on the sex chromosomes. Taken together, our results show that H3K79me2 positively correlates with male germ cell genetic program throughout spermatogenesis, with DOT1L predominantly inhibiting rather than activating gene expression. Interestingly, while DOT1L appears to directly regulate the (re)activation of X genes following meiotic sex chromosome inactivation, it also controls the timely expression of (autosomal) differentiation genes during spermatogenesis.

DOT1L (*Disruptor of telomeric silencing 1-like*) is the sole H3K79 (lysine 79 of histone H3) methyltransferase[1] and is highly conserved in eukaryotes. It has been shown to be involved in multiple biological processes, such as embryonic development, cell cycle progression, cell reprogramming, differentiation, and proliferation, as well as in the emergence of mixed lineage leukemia[2–7].

Mechanistically, DOT1L protein interacts with the active form of RNA polymerase II[8] and with protein complexes that promote transcription elongation, such as EAP (for ENL-associated proteins) and SEC (for Super Elongation Complex)[9,10]. These interactions are at the basis of DOT1L involvement in cancer development, as in some types of mixed lineage leukemia in which the fusion between MLL1 protein and DOT1L partners (such as ENL, AF9, or AF10) leads to the ectopic activation of MLL1 target genes by DOT1L[11–13]. More recently, DOT1L has also been found to interact with the transcription initiation factor TFIID[14]. All forms of H3K79 methylation (mono, di, and trimethylation) are enriched at the promoter and gene body of transcriptionally active genes leading to the view that DOT1L acts as a transcriptional activator[15–17]. Yet, recent molecular studies have challenged this model by reporting that DOT1L loss of function does not lead to extensive gene downregulation, as one could have expected, but rather leads to the de-repression of hundreds to thousands of genes, depending on the cell type (for review see Wille and Sridharan[6]).

DOT1L expression is ubiquitous, with a variable level across tissues and cell types, but is remarkably high in the testis, during spermatogenesis, and more particularly in meiotic and post-meiotic male germ cells[18–21]. We and others have recently shown that DOT1L is essential for the post-meiotic

[1]Université Paris Cité, CNRS, Inserm, Institut Cochin, F-75014 Paris, France. [2]Université Paris Cité, CEA, Stabilité Génétique Cellules Souches et Radiations, Fontenay-aux-Roses Paris, France. [3]Laboratoire des Cellules Souches Germinales, Université Paris-Saclay, CEA, Stabilité Génétique Cellules Souches et Radiations, Fontenay-aux-Roses Paris, France. [4]Departament de Biologia Cel·lular, Fisiologia i Immunologia, Universitat Autònoma de Barcelona, Cerdanyola del Vallès Barcelona, Spain. [5]Genome Integrity and Instability Group, Institut de Biotecnologia i Biomedicina, Universitat Autònoma de Barcelona, Cerdanyola del Vallès Barcelona, Spain. [6]MOABI, plateforme de Bioinformatique de l'APHP, Paris, France. [7]Sorbonne Université, APHP Hôpital Pitié-Salpêtrière, Paris Brain Institute—ICM, INSERM U1127, CNRS, UMR 7225 Paris, France. [8]These authors contributed equally: Julie Cocquet, Laïla El Khattabi. ✉e-mail: julie.cocquet@inserm.fr; laila.el-khattabi@aphp.fr

differentiation of spermatids into functional spermatozoa, as well as for spermatogonia self-renewal[18,22–24]; yet, its role in gene expression regulation in male germ cells remains unclear. Importantly, spermatogenesis is one of the biological processes during which transcription and chromatin remodeling are the most dynamic[25–27]. It consists of very specific cell stages, i.e. spermatogonial stem cell stage, spermatogonia proliferation, meiosis, and spermatid differentiation (see Fig. 1a), which are defined by distinct genetic programs[25–29]. Spermatogenesis is also characterized by extensive chromatin reorganization that is tightly associated with gene expression dynamics. During meiosis, homologous chromosomes condense, pair, and recombine, while heterologous sex chromosomes are temporarily silenced. Later, towards the end of spermatogenesis, most histones replaced by protamines resulting in a tight compaction of the sperm chromatin and global transcription shut down[30–34]. Here, we took advantage of these unique features to provide a comprehensive view of the role of DOT1L in gene expression, in particular in the context of H3K79me2 dynamics, throughout spermatogenesis.

## Results

### H3K79me2 at gene body and enhancers correlates with active transcription during male germ cell differentiation

H3K79 methylation is known to correlate with gene activity as it marks the transcriptional start site (TSS) and gene body of transcribed genes[5,16]. Specifically during spermatogenesis, H3K79me2 has been shown to increase at meiosis and before histone-to-protamine transition[19–21,35], suggesting a role for this mark in these processes. Hence, we focused on H3K79me2 (instead of H3K79me1 or H3K79me3) because it is both dynamic and abundant in male germ cells, and it is the most studied mark in other cell types, facilitating the comparison of results. We performed H3K79me2 ChIP-seq experiments on four representative timepoints of male germ cell development: (i) primary culture of germinal stem cells (GSC) accounting for premeiotic stage, (ii) primary spermatocytes (SCI) at the meiotic stage, and (iii) round spermatids (RS) and (iv) elongating spermatids/condensed spermatids (ES) both representing postmeiotic stages (Fig. 1a). Experimental replicates showed strong correlation for each cell stage (Supplementary Fig. 1a). The number of peaks between replicates was also very similar (Supplementary Fig. 1b), except for ES. This discrepancy is likely due to the transition of ES cells from a nucleosomal structure to a protamine-organized chromatin, which increases cell population variability. We, therefore, decided to exclude ES from subsequent analyses.

Genomic annotation showed that most H3K79me2 peaks are located at promoters [63% in GSC ($n = 10{,}089$), 55% in SCI ($n = 19{,}480$) and 45% in RS ($n = 25{,}555$)], and intragenic regions [35% in GSC ($n = 5621$), 39% in SCI ($n = 13{,}880$) and 37% in RS ($n = 21{,}348$)], while a smaller number [0.1% in GSC ($n = 15$), 0.6% in SCI ($n = 219$) and 1% in RS ($n = 535$)], was found to map at downstream regions (Fig. 1b). This distribution was significantly different between cell types (Pearson's $\chi^2$: $p < 2.2e{-}16$). In particular, we observed a remodeling of the H3K79me2 genomic landscape, with a progressive increase of the proportion of peaks at distal intergenic regions throughout differentiation [1% in GSC ($n = 170$), 5% in SCI ($n = 2122$) and 15% of peaks in RS ($n = 8788$), Fig. 1b and Supplementary Fig. 1b]. At the gene level, we detected the strongest H3K79me2 enrichment right after the transcriptional start site (TSS ± 1.5 kb), which slowly decreases in intensity along the gene body, in all cell types (Fig. 1c), consistently with what was previously described in somatic cells[5]. Moreover, we found a positive correlation between H3K79me2 enrichment and gene expression in all cell types (ANOVA: $p < 2.2e{-}16$), indicating that the higher the H3K79me2 enrichment at a gene, the more it is expressed (Fig. 1c).

A detailed analysis of H3K79me2 dynamics between cell types revealed a high conservation of peaks present in GSC throughout spermatogenesis, with ~54% of GSC peaks maintained in SCI and RS. In contrast, a high number of peaks (~26% of the total number of peaks) were de novo established at meiosis in SCI and even greater (~48%) after meiosis, in RS (Fig. 1d and Supplementary Data 1). Interestingly, H3K79me2 regions specifically acquired at meiotic or postmeiotic stages ("SCI-RS" and "RS-

spe" peaks) were enriched in genes involved in spermiogenesis, such as spermatid development, and cilium assembly, organization, and movement, while peaks conserved throughout spermatogenesis ("common" peaks) mapped to genes of various pathways, including Wnt signaling and actin filament organization pathways (Fig. 1e). The analysis of H3K79me2 genomic distribution among cell types, revealed a high proportion of peaks mapping to distal intergenic regions among those that were specifically established at the postmeiotic stage (RS-spe), while peaks common to all stages were almost exclusively enriched at gene promoters (Fig. 1f).

To better characterize H3K79me2 peaks, we investigated chromatin dynamics during spermatogenesis using six different histone marks (H3K4me1, H3K4me3, H3K27ac, H3K27me3, H3K36me3, and H3K9me3). We employed the ChromHMM tool[36] to define 18 chromatin states characterized by distinct combinations of histone post-translational modifications (PTMs), which were then merged into 13 chromatin states to simplify downstream analysis (see Fig. 2a). When comparing the different chromatin states with H3K79me2-enriched regions ("H3K79me2+") or those devoid of H3K79me2 ("H3K79me2−"), we found that H3K79me2 was enriched at transcription and enhancer regions, with differences between cell stages (Fig. 2b and Supplementary Data 1). In particular, we observed H3K79me2 located at weak transcription-related chromatin state (TxWk) for GSC (28% of H3K79me2+ vs. 3% of H3K79me2−; $\chi^2$, $p = 2e{-}46$) and at bivalent TSS for RS (9% vs. 1%; $\chi^2$, $p = 3e{-}15$). At both SCI and RS stages, H3K79me2+ regions were also over-represented in chromatin states flanking the TSS, compared to H3K79me2− regions [5% vs. 0.5% in SCI ($\chi^2$, $p = 2e{-}10$) and 6% vs. 0.6% in RS ($\chi^2$, $p = 4e{-}12$)]. Importantly, the distribution of H3K79me2 at enhancers changed throughout male germ cell differentiation. Specifically, in GSC, compared to H3K79me2-regions, H3K79me2+ ones were enriched in active enhancers (2.8% of H3K79me2+ regions vs. 0.4% of H3K79me2−; $\chi^2$, $p = 5.8e{-}5$) and weak enhancers (12.3% vs. 6.2%, $\chi^2$, $p = 0.0113$). In contrast, in SCI and RS, H3K79me2+ regions were highly enriched in genic enhancers [49.9% vs. 4% in SCI ($\chi^2$, $p = 3e{-}120$) and 36% vs. 3.7% in RS ($\chi^2$, $p = 1.6e{-}64$)] (Fig. 2b).

We next investigated the dynamic of chromatin state transitions with regard to H3K79me2 throughout spermatogenesis, from GSC to RS stages (Fig. 2c, Supplementary Fig. 2, and Supplementary Data 1). Priori to meiosis, in GSCs, most H3K79me2+ chromatin states (91%) were categorized as weak (i.e. TxWk and EnhW) or quiescent (Quies). In contrast, in meiotic cells (SCI), H3K79me2 was more associated with active (i.e. Tx and EnhG) than weak chromatin states (60% vs. 21%). Interestingly, after meiosis, in RS, we observed a higher diversity of H3K79me2+ chromatin states, with 34% found in active, 30% in weak, 17% in bivalent, 17% in quiescent and 2% in heterochromatin states (Fig. 2c). The H3K79me2 mark was highly enriched (89%) in RS bivalent regions (i.e. TssBiv and EnhBiv) (Fig. 2b). Further analysis revealed that 70% of H3K79me2+ bivalent regions specifically acquired H3K79me2 mark at RS stage (i.e. they were devoid of H3K79me2 in GSC and SCI) (Supplementary Fig. 2a). Among the H3K79me2+ bivalent regions in RS, ~64% newly acquired the H3K27me3 mark at this stage (these regions mapped predominantly to TSSFlnk (35%) or Quiescent (16%) chromatin states in SCI) (Fig. 2c and Supplementary Fig. 2a). As for H3K79me2+ active chromatin states (i.e. TSSA, Tx, EnhG, or EnhA), we observed a switch from weak (TxWk or EnhW) to active states (Tx or EnhG) at the entry in meiosis. Indeed, half of weak H3K79me2+ regions in GSC become active in SCI (i.e. EnhG or Tx regions), and most of these regions are maintained after meiosis in RS (Fig. 2c and Supplementary Fig. 2b).

Finally, with the ChromHMM model, we defined four categories of enhancers: (i) active enhancers (EnhA) characterized by the presence of H3K4me1 and H3K27ac, (ii) weak enhancers (EnhW) characterized by H3K4me1 (with low H3K27ac level), (iii) bivalent enhancers (EnhBiv) enriched in H3K4me1, H3K27me3 and, to a lesser extent, H3K27ac, and (iv) genic enhancers (EnhG), which are enriched in H3K4me1, H3K36me3 and H3K27ac (and to a lesser extent H3K27me3). For all types of enhancers, we found that enrichment in H3K79me2 is associated with significantly higher expression levels of neighboring genes

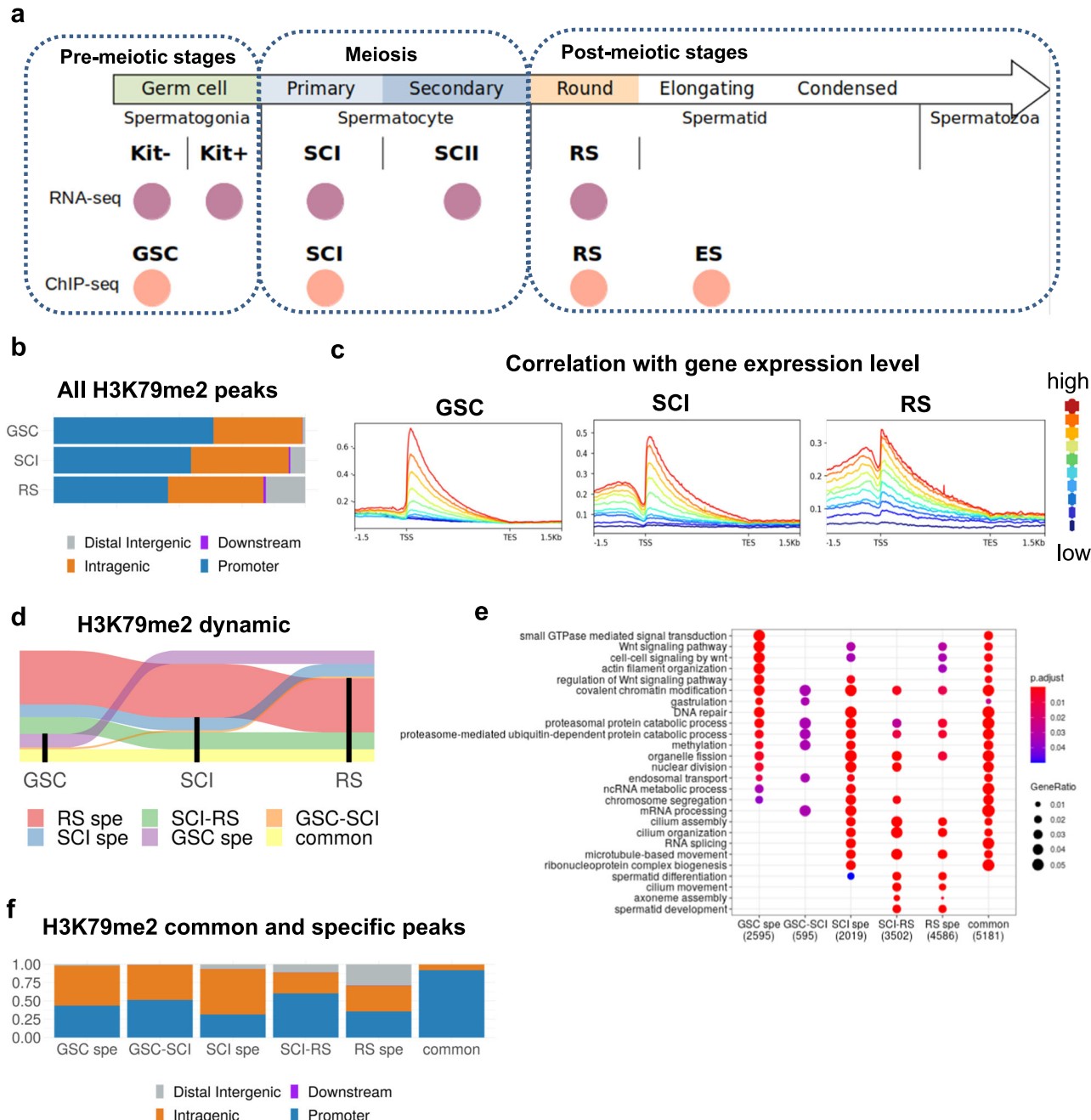

**Fig. 1 | Characterization of H3K79me2 dynamic during male germ cell differentiation. a** List of the datasets used in the present study: RNA-seq analyses were performed on *Dot1l*-KO vs. CTL samples and H3K79me2 ChIP-seq on WT samples. Kit− and Kit+ correspond, respectively, to undifferentiated and differentiated spermatogonia, GSC to germinal stem cells (i.e., in culture amplified kit− cells), SCI and SCII to primary and secondary spermatocytes, and RS to round spermatids (RS). **b** Genomic annotation of H3K79me2 ChIP-Seq peaks in GSC, SCI, and RS, using ChIPSeeker. **c** H3K79me2 enrichment according to the gene expression level in GSC, SCI, and RS. Blue: low expressed genes, red: high expressed genes. Gene expression level is divided into percentile groups in each cell type. **d** Alluvial plot showing the dynamic of H3K79me2 peaks throughout spermatogenesis. Regions enriched in H3K79me2 in each cell type (GSC, SCI, or RS) are shown in black. Only groups containing more than 100 regions are shown. **e** Ontology of the genes enriched in H3K79me2 at specific stages using clusterProfiler, *p*-value is adjusted according to Benjamini–Hochberg correction and GeneRatio corresponds to the ratio of deregulated genes among all genes in the pathway. **f** Genomic annotation of H3K79me2 ChIP-Seq peaks that are stage-specific (GSC spe, SCI spe, or RS spe), or common to two successive stages (GSC-SCI or SCI-RS) or those common to all stages (common).

compared to enhancers devoid of H3K79me2 (Fig. 2d, Supplementary Fig. 3).

When considering the dynamics of the 3D genome remodeling, we detected that H3K79me2 was positively associated with open chromatin genomic compartments (A-compartments) in SCI and RS (permutation test based on 10,000 permutations, normalized *z*-score −0.05, *p* < 0.05) (Fig. 2e and f). Moreover, we found an association between H3K79me2 and the genomic position of different structural meiotic features such as meiotic cohesins (RAD21L and REC8) in SCI and RS (permutation test based on 10,000 permutations, normalized *z*-score −0.05, *p* < 0.05). Importantly, both meiotic cohesins were enriched in TSSBiv in SCI and RS, reinforcing the role of cohesins in germline transcription (Fig. 2e and f).

In summary, H3K79me2 peaks are associated with active chromatin states and compartments, as well as with meiotic cohesins. When located at

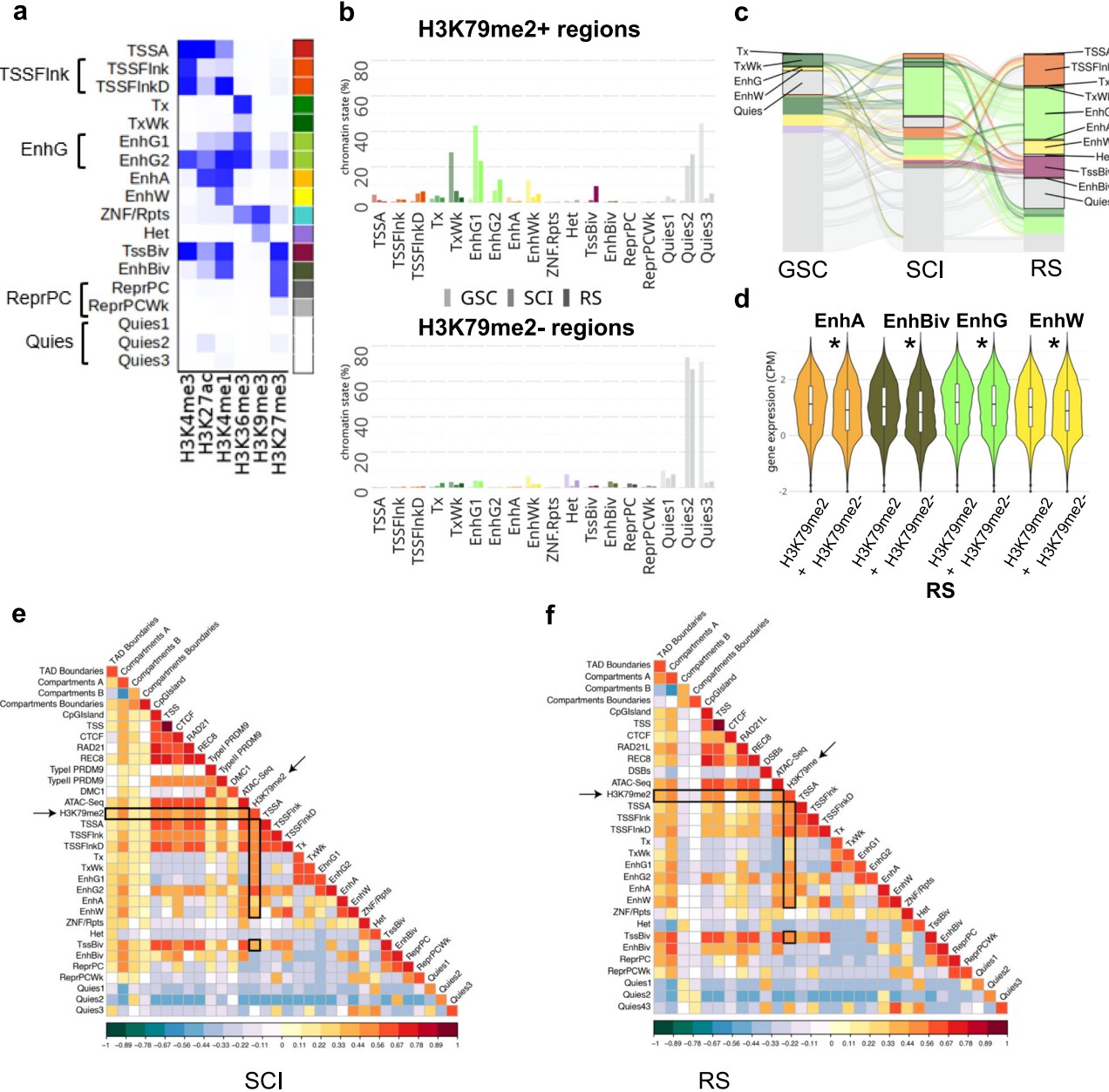

**Fig. 2 | Chromatin environment at H3K79me2 regions in male germ cells. a** The 18 chromatin states model obtained with ChromHMM. TSSA Active Transcription start site, TSSFlnk flanking TSS, and TSSFlnkD flanking TSS downstream (which were pooled in the state named TSSFlnk), Tx strong transcription, TxWk weak transcription, EnhG1 and EnhG2 genic enhancers (which were pooled in one state named EnhG), EnhA active enhancer, EnhWk weak enhancer, ZNF/Rpts ZNF genes and repeats, Het Heterochromatin, TSSBiv bivalent/poised TSS, EnhBiv bivalent enhancer, ReprPC repressed PolyComb and ReprPCWk weak repressed polycomb (which were pooled in one state named ReprPC), Quies1, Quies2, and Quies3 Quiescent/Low (which were pooled in one state called Quies). **b** Distribution of chromatin states among H3K79me2 enriched regions (H3K79me2+) and H3K79me2 devoid regions (H3K79me2−). **c** Alluvial plot showing the dynamic of

H3K79me2+ chromatin states (as defined with ChromHMM) throughout spermatogenesis. Only groups containing more than 100 regions are shown. **d** Mean gene expression level (log(cpm)) of genes associated with enhancers enriched (+) or not (−) in H3K79me2, in RS. Stars indicate *p*-values calculated using the Wilcoxon test (**p* < 0.05). **e** and **f** Heatmaps obtained by regioneR (multicomparison) displaying correlations between H3K79me2 in SCI (**e**) and in RS (**f**) and TAD boundaries, A and B compartments, compartment switch (from A to B and vice versa), CpG islands, CTCF, cohesins (RAD21L and REC8), ATAC-Seq, transcription start sites (TSS) and the 18 chromatin states defined in Fig. 2a (and indicated as SC1–SC18 for primary spermatocytes and RS1–RS18 for round spermatids). SCI comparison also included PRDM9 sites (Type I and II) and DMC1 sites. RS included post-meiotic DSBs.

enhancers, these peaks are associated with higher expression levels of nearby genes, regardless of the enhancer type (genic, non-genic active, weak, or bivalent). Thus, H3K79me2 at gene bodies and distal enhancers serves as a marker of active transcription. Our analyses also reveal a shift in the H3K79me2 landscape at the mitosis-to-meiosis transition, characterized by a change from weak chromatin states to active ones, and a particularly high number of peaks acquired post-meiotically, especially at distal intergenic and bivalent regions. This gain of H3K79me2 peaks in RS is accompanied by

a gain of the H3K27me3 mark consistent with H3K79me2's role in marking bivalent regions in RS.

**Spike-in RNA-seq analyses reveal that DOT1L has a repressive effect on gene expression**

In order to characterize the role of DOT1L in transcriptional regulation during male germ cell differentiation, we performed RNAseq analyses from a *Dot1l* male germline-conditional KO mouse model (described previously

in Blanco et al.[18]). Male germ cells from five different stages were investigated: undifferentiated spermatogonia (Kit-), differentiated spermatogonia (Kit+), SCI, SCII and RS (Figs. 1a, 3a, Supplementary Figs. 4, 5, and Supplementary Data 2). Known markers of each cell type confirmed the expected expression pattern (Supplementary Fig. 4c).

To investigate the existence of a global effect of *Dot1l*-KO on the transcriptome, we included spike-in sequences to normalize gene expression levels to the number of cells from which RNA was extracted[37]. Differential gene expression analyses showed that gene deregulation intensifies as spermatogenesis progresses, with 140 deregulated genes in premeiotic spermatogonial stages and 925 deregulated genes in meiotic and postmeiotic cells ($p < 0.05$; FDR < 0.05; FC > 1.5, Fig. 3b). We also observed a bias towards gene upregulation in KO samples at all stages of male germ cell development (Fig. 3c), which is surprising given the positive correlation between H3K79 methylation and transcription. This observation was confirmed by studying the distribution of gene expression levels in meiotic and postmeiotic cells: the average expression of autosomal genes was significantly higher in SCI, SCII, and RS in KO vs. CTL (Fig. 3d). The "classic" (without spike-in) normalization method[18] revealed a high overlap of deregulated genes (i.e. 94%, 96%, and 77% for SCI, SCII, and RS, respectively), while the spike-in normalization method identified a higher number of deregulated genes.

We next examined the dynamic of deregulated genes and observed distinct programs before and after meiosis. On one hand, most deregulated genes in Kit+ were not deregulated in meiotic and postmeiotic cells, and reciprocally. On the other hand, 31% of deregulated genes ($n = 290$) were common between the meiotic and postmeiotic stages (Fig. 3e).

Further characterization of the length of deregulated genes revealed that upregulated genes in SCI, SCII, and RS are often long genes ($\chi^2$, $p < 0.05$ between upregulated and all genes, for the 90th percentile of size in SCI, SCII, and RS), while downregulated genes in Kit+ cells tend to be shorter ($\chi^2$, $p = 0.00025$ between downregulated and all genes, for "spike-in" analyses) (Supplementary Fig. 4d). The functional annotation of deregulated genes was overall similar to that of other genes, with a majority of "protein coding" genes in all categories and cell types. However, there was a significant increase in the proportion of lincRNA and lncRNA among the downregulated genes, particularly visible in Kit+ cells ($\chi^2$ between downregulated and all genes, $p = 3.61e{-}06$ for lincRNA; $p = 0.0029$ for lncRNA) (Supplementary Fig. 4e).

Altogether, the extent of gene deregulation induced by *Dot1l* loss increases progressively with male germ cell development. At all stages, gene deregulation is unexpectedly biased towards upregulation and involves different sets of genes between pre-meiotic, meiotic, and post-meiotic stages.

## DOT1L has opposite effects on gene expression depending on H3K79me2 but also on H3K27me3 chromatin environment

To address the link between gene expression deregulation following *Dot1l* loss and H3K79 methylation, we compared the H3K79me2 enrichment pattern at deregulated and not deregulated genes. We observed that H3K79me2 is enriched at the body of the majority of not deregulated genes, and of downregulated genes (in RS) while at least 80% of upregulated genes are devoid of the mark in both SCI and RS ($\chi^2$: $p < 2.2e{-}16$ for down vs. not genes in SCI and RS) (Fig. 4a and Supplementary Data 3). Since we found a positive correlation between H3K79me2 enrichment at enhancers and the expression level of neighboring genes (Fig. 2d), we investigated whether the deregulated genes in *Dot1l*-KO male germ cells are associated with H3K79me2+ enhancers (Fig. 4b). The results show that, in SCI and RS, the enrichment pattern between downregulated genes and unaffected genes is similar, with a majority of genes enriched in H3K79me2 at their body and/or nearby enhancers (presence of H3K79me2 at gene body and/or enhancers in ~54% of downregulated genes and ~79% of unaffected genes for SCI; ~91% of downregulated genes and ~90% of unaffected genes for RS). In contrast, upregulated genes are significantly enriched in H3K79me2 enhancer-only regions (35% among upregulated genes vs. 24% of unaffected genes in SCI, and 40% vs. 24% in RS, respectively; $\chi^2$: $p < 2.2e{-}16$) (Fig. 4b).

Upregulated genes are, therefore, mostly either devoid of H3K79me2 or enriched in H3K79me2 strictly at nearby enhancers and rarely at their gene body. These results indicate different mechanisms of control of gene expression by DOT1L, and that the repressive effect of DOT1L on gene expression is, at least partially, independent of H3K79me2.

To search for what drives the observed gene deregulation in an H3K79me2-independent manner, we next examined the chromatin environment of deregulated genes using ChromHMM (Fig. 2a) to assign the predominant chromatin state flanking the promoter of deregulated genes. First, we analyzed downregulated genes and did not observe any specific chromatin signature different from that of non-deregulated genes, except for a decrease in the proportion of "genic enhancers" chromatin state in SCI and RS (EnhG, $\chi^2$: $p = 0.0006$ for down vs. not genes in SCI and $p = 0.0025$ in RS) and an increase in "TSS" chromatin states (i.e. which have high levels of H3K4me3; TSSFlnk, $\chi^2$: $p = 6.01e{-}17$, and TssBiv, $\chi^2$: $p = 0.009$) for RS (Fig. 4c and Supplementary Data 4).

In contrast, upregulated genes were highly associated with bivalent enhancers in RS (EnhBiv: $p = 0.25$ for SCI and $p = 0.00078$ for RS), meiotic cohesins (Rad21L and REC8) and A-compartments (Fig. 4c and d). Upregulated genes were also associated with repressed Polycomb (ReprPC: $p = 3.6e{-}05$ for SCI and $2.8e{-}28$ for RS) and quiescent (Quies: $p = 7.01e{-}17$ for SCI and $p = 7.27e{-}30$ for RS) chromatin states (Fig. 4c). When including all chromatin states enriched in H3K27me3 (i.e. EnhBiv, ReprPC, and TSSBiv), we found that ~30% and ~48% of upregulated genes are in H3K27me3-rich regions in SCI and RS, respectively, compared to 18% and 32% for not deregulated genes. Conversely, only ~2% and ~4% of upregulated genes are in H3K27ac-rich regions in SCI and RS, respectively, compared to 34% and 37% for not deregulated genes.

These results indicate that many upregulated genes are located in H3K27me3-rich/H3K27ac-poor chromatin states that are accessible and associated with meiotic cohesins. The H3K27me3 enrichment profile of deregulated genes confirmed that upregulated genes are more enriched in H3K27me3, while downregulated genes are rather enriched in H3K4me3, at their promoters (Fig. 4e).

To test the accuracy of this model, we next performed quantitative (spike-in) CUT&Tag experiments for H3K27ac and H3K27me3 in *Dot1l*-KO and control RS, in triplicates (Supplementary Fig. 1c, Fig. 4f and g). Normalized enrichment profiles showed that in control RS, genes upregulated in the KO have low levels of H3K27ac and very high levels of H3K27me3 at their promoter/TSS. Upon *Dot1l* loss, their enrichment in H3K27me3 decreases while H3K27ac slightly increases, consistent with their upregulation. Conversely, in control RS, downregulated genes have higher levels of H3K27ac and are almost devoid of H3K27me3. Upon *Dot1l* loss, their H3K27ac signal is visibly reduced while H3K27me3 remains unchanged, consistent with their downregulation. Finally, genes unaffected by *Dot1l*-KO have intermediate H3K27me3 levels (lower than upregulated genes) and low H3K27ac enrichment at their promoter/TSS (Fig. 4f and g). Altogether, these findings confirm the model derived from the ChromHMM analyses and show that the impact of DOT1L on gene expression depends on the local chromatin environment of these genes, particularly on H3K27me3, H3K27ac, and/or H3K79me2 enrichment (see Fig. 6).

Many other upregulated genes do not bear any specific chromatin signature since they are located in regions annotated as quiescent, i.e. without any of the above-described marks. We investigated whether their upregulation could result from the downregulation of transcriptional repressors. For this, we listed downregulated genes annotated as repressors and enriched in H3K79me2, which presumably direct the expression of DOT1L-target genes. In contrast to previous observations in B and T cells[38], we did not find a downregulation of the gene encoding for the H3K27me3 methyltransferase EZH2 in *Dot1l*-KO male germ cells at any cell stage. However, our "spike-in" RNA-seq analysis confirmed the downregulation of *Bcl6* (previously observed in Blanco et al.[18]) in meiotic and postmeiotic cells and additionally found the downregulation of *Bcorl1*, which encodes a BCL6 co-repressor, in SCII and RS. Other genes classified as repressors were

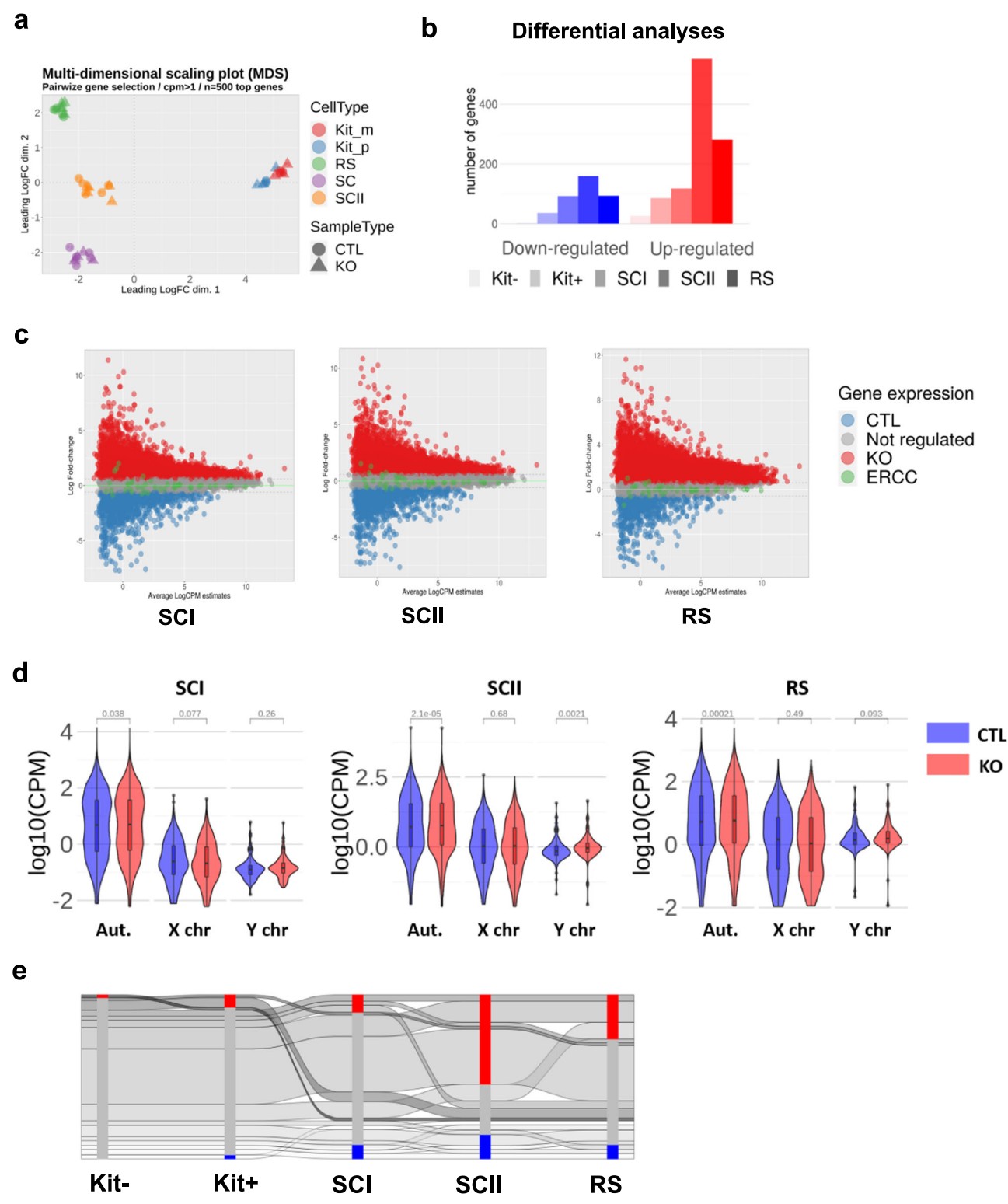

**Fig. 3 | Transcriptional changes throughout spermatogenesis upon *Dot1l* knock-out. a** Multi-dimensional scaling plot showing RNA-Seq data from two genotypes (*Dot1l*-KO and CTL cells) and five male germ cell types (*n* = 3 KO and 3 CTL for Kit- and Kit+; *n* = 5 KO and 5 CTL for SCI, SCII and RS including 2 non-*spike in* data. NB. SCI, SCII, and RS data have already been published in Blanco et al.[18]): Kit− undifferentiated spermatogonia, Kit+ differentiated spermatogonia expressing the c-kit marker, SCI primary spermatocytes, SCII secondary spermatocytes, and RS round spermatids. **b** Number of deregulated genes according to a differential expression including a spike-in normalization (*n* = 3 replicates for all cell types and conditions) in five male germ cell types (*p* < 0.05; FDR < 0.05; FC > 1.5). **c** MD plot representing the number of genes associated with CTL and KO samples centered on spike-in value. CTL: genes more associated with the control condition, KO: genes more associated with the KO condition, ERCC: *spike-in* data (NB: MD plots without ERCC normalization were already shown in Blanco et al.[18]). **d** Log10(CPM) expression of all genes in CTL or *Dot1l*-KO (KO) sample expression in SCI, SCII, and RS. Adjusted *p*-values using the Wilcoxon test adjusted with Benjamini–Hochberg correction are indicated above each KO vs. CTL comparison. **e** Dynamics of the genes deregulated in *Dot1l*-KO vs. CTL male germ cells from premeiotic to post-meiotic stages (spike-in RNA-Seq analysis).

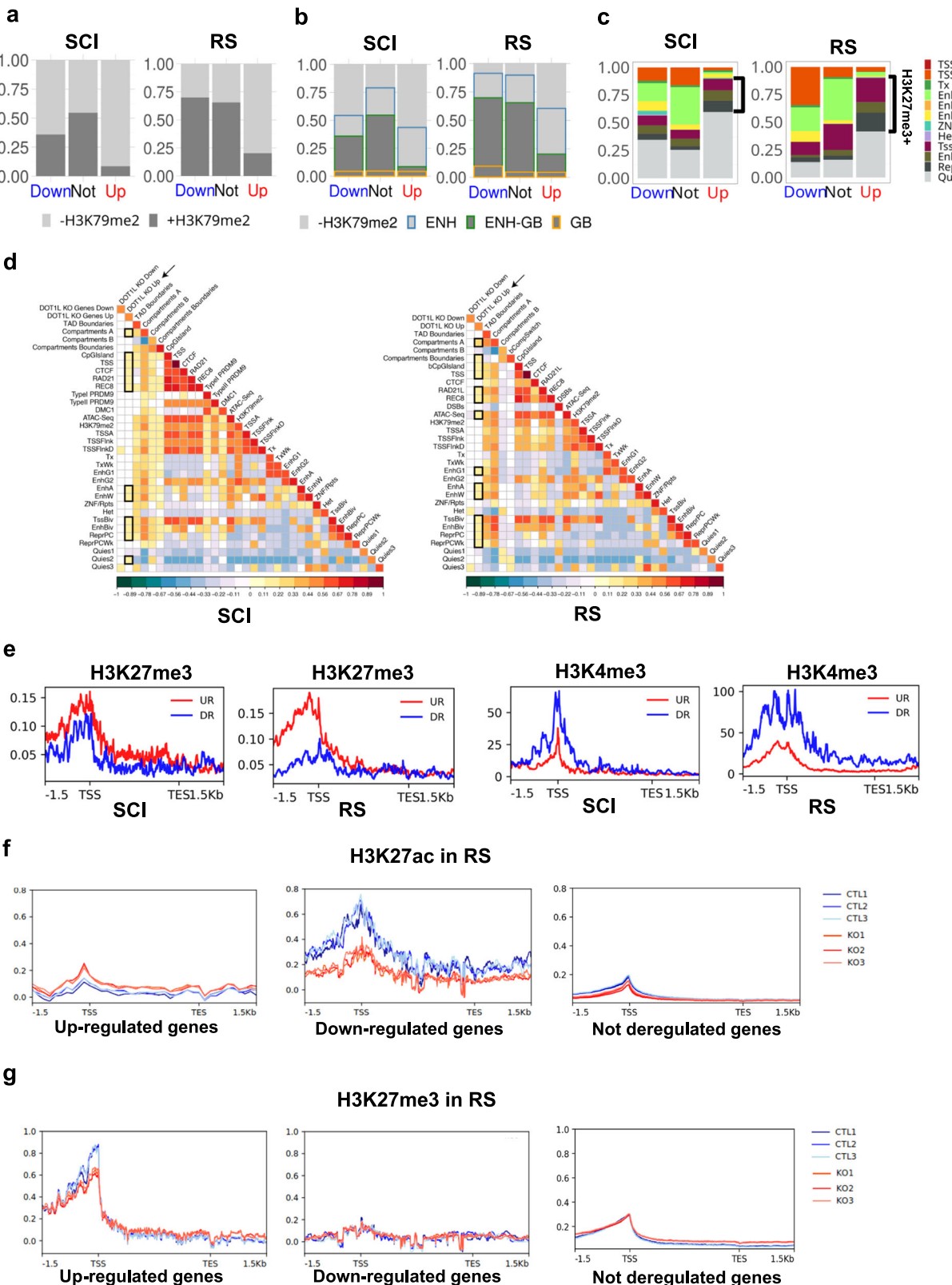

found in this list, such as *Foxo4*, *Iqsec2*, *Rps6ka3*, *Stk26*, *Zfp36l3*, and *Zkscan17* (Supplementary Data 2). Interestingly, we also noted the down-regulation of *Hdac8*, encoding for a histone deacetylase previously shown to deacetylate H3K27 residue[39,40]. In premeiotic Kit− or Kit+ spermatogonial cells, none of these repressor genes were downregulated.

Collectively, our data indicate that, in the male germline, DOT1L represses gene expression via several mechanisms, linked to the chromatin environment at both promoters and nearby regulatory elements, as well as through the downregulation of transcriptional repressors.

**Fig. 4 | Correlation between differentially expressed genes and their chromatin environment. a** Distribution of H3K79me2 enriched (H3K79me2+, dark gray) or not (H3K79me2−, light gray) genes among different groups of differentially expressed genes found by spike-in RNA-seq analysis of *Dot1l*-KO vs. CTL samples (Up: upregulated genes, Down: downregulated genes, Not: not deregulated genes) following *Dot1l* KO. H3K79me2 enrichment was evaluated at the gene body in SCI and RS. Kit−/GSC data was not included because only a few genes were found deregulated at this stage. (NB. RS barplot was already shown in ref. 19 but without ERCC normalization). **b** Distribution of H3K79me2 enrichment at or nearby genes [at the gene body (GB), at enhancers/regulatory elements (ENH) as defined by ChromHMM analysis, at both regions (GB-ENH) or in none of these (−H3K79me2)], in different gene categories (up, down or not deregulated), in SCI and RS. **c** Predominant chromatin states (as defined in Fig. 2a) associated with the promoter region (±3 kb interval around the TSS) of deregulated genes or not

deregulated genes, in SCI and RS. The bracket indicates H3K27me3+ chromatin states. **d** Heatmaps obtained by regioneR (multicomparison) displaying correlations between deregulated genes in SCI (left) and in RS (right) and TAD boundaries, A and B compartments, compartment switch (from A to B and vice versa), CpG islands, CTCF, cohesins (RAD21L and REC8), ATAC-Seq, transcription start sites (TSS) and the 18 chromatin states defined in Fig. 2a (and indicated as SC1–SC18 for primary spermatocytes and RS1–RS18 for round spermatids). Primary spermatocytes also included PRDM9 sites (Type I and II) and DMC1 sites. Round spermatids included post-meiotic DSBs. **e** H3K27me3 and H3K4me3-enrichment profiles at differentially expressed genes in wild-type SCI and RS from published ChIP-seq data. **f** and **g** H3K27ac and H3K27me3 enrichment profiles at upregulated, downregulated, or not deregulated genes from spike-in CUT&Tag experiments performed on 3 *Dot1l*-KO (KO) and 3 control (CTL) RS samples.

## DOT1L activates X gene expression following meiotic sex chromosome inactivation, in correlation with a specific H3K79me2 dynamic

Most repressor genes that are downregulated in *Dot1l*-KO are encoded by the X chromosome. We, therefore, examined the chromosome location of deregulated genes and found an over-representation of X-encoded genes among downregulated genes in *Dot1l*-KO SCII and RS, reaching 47% of downregulated genes in RS (Fig. 5a and Supplementary Fig. 6a). When we measured the mean expression level for all autosomal, X-linked, and Y-linked genes in CTL and KO cells (Fig. 3d), we observed very low levels for X- and Y-linked genes in KO and CTL SCI, corresponding to the transcriptional shutdown of XY gene expression at the pachytene stage, a phenomenon termed meiotic sex chromatin inactivation (MSCI)[41]. No significant difference in gene expression level was observed between KO and CTL samples for X-linked genes, indicating that the over-representation of X-linked genes among downregulated genes in SCII and RS applies to a subset of X-linked genes (Fig. 3d).

As a result of both the downregulation of X-linked genes and the upregulation of autosomal genes, the average Log2 fold change between KO and CTL was significantly different for autosomal versus X-encoded deregulated genes (Fig. 5b, Supplementary Fig. 6b). Interestingly, many downregulated genes co-localized with cohesin rich regions (REC8 and RAD21L) on the X chromosome (Fig. 5c). Surprisingly, chromosome Y-linked gene expression in *Dot1l*-KO displayed a different profile with a slight but significant upregulation in SCII (Fig. 3d). By RNA-seq, however, few to no Y-linked genes were found significantly deregulated, except *Zfy1*, found downregulated and pseudogenes corresponding to the multicopy genes *Ssty* and *Sly*, which were upregulated in SCII (Supplementary Data 2). By qRT-PCR performed in RS, we confirmed the down-regulation of *Zfy1* and the upregulation of *Sly1* but not of *Ssty1* or *Ssty2* (Supplementary Fig. 6c). *Slx* and *Slxl1*, the X-linked homologs of *Sly*, were not deregulated while the downregulation of the X-encoded single copy genes *Ube2a* and *Hdac8* was confirmed. It is worth noting that *Sly* and *Ssty* are present in >100 copies and represent the vast majority of Y-linked genes. They are only expressed in SCII and RS; their upregulation could, therefore, explain the global upregulation of Y gene expression observed in SCII and in RS (Figs. 3d, 5b, and Supplementary Fig. 6b). These observations point to a complex role of DOT1L in the regulation of Y chromosome-encoded genes.

The observed bias towards the deregulation of XY genes could be related to their specific chromatin environment in male germ cells as a consequence of MSCI. Indeed, in SCI, the transcriptional repression of sex chromosomes is associated with an enrichment of the heterochromatin mark H3K9me3 along the sex chromosomes[42]. As primary spermatocytes develop into secondary spermatocytes and then round spermatids, XY gene expression is activated with the sex chromosomes acquiring active histone marks, such as H3K4me3 and H3K27ac. However, the average gene expression level of XY-encoded genes remains lower than that of autosomal genes, attributed to the maintenance of an H3K9me3-rich chromatin environment[29,43].

Using our ChIP-seq data, we explored the H3K79me2 dynamic during wild-type spermatogenesis and observed a distinct enrichment pattern between sex chromosomes and autosomes (after normalization to the chromosome size). Specifically, H3K79me2 enrichment on sex chromosomes increases dramatically from SCI to the RS stage (Fig. 5d), with many of the peaks acquired at the RS stage (i.e. ~25%) located on these chromosomes.

This dynamic was directly linked with the observed downregulation of X genes, as most X downregulated genes were marked by H3K79me2 in wild-type RS (Fig. 5e). Furthermore, we observed that ~60% of down-regulated genes are located in de novo acquired peaks (i.e. RS-specific peaks) vs. only 35% of non-deregulated genes. By ChIP-qPCR in RS, we observed that H3K79me2 enrichment is lost at the body of several downregulated X-encoded genes and of *Zfy1*, upon *Dot1l* loss (Fig. 5f), indicating that DOT1L may regulate the activation of X-linked genes following MSCI via H3K79me2. When compared to other histone marks known to be differentially enriched between sex chromosomes and autosomes in RS[29,43], H3K79me2 appeared to be the most dynamic mark between SCI and RS, with a 4-fold increase for chromosome Y and a 24-fold increase for chromosome X (Supplementary Figs. 6d, e and 7a). This makes H3K79me2 more enriched than H3K27ac, an active mark previously described to promote X gene activation in RS[43,44]. By quantitative CUT&Tag, we confirmed that X-linked genes are enriched in H3K27ac at their promoter/TSS, and observed a decrease in H3K27ac upon *Dot1l* loss, consistent with their downregulation (Supplementary Fig. 7b).

When including GSC datasets, it became clear that chromatin remodeling dynamics differ between the X and Y chromosomes before meiosis. For instance, H3K79me2 enrichment, normalized to the chromosome size, was higher on the Y than on the X, at the GSC to SCI transition; this difference was even more striking for H3K4me1, H3K27ac, and H3K27me3 (Supplementary Fig. 7a). As for H3K9me3, we previously showed that it is similarly enriched on the Y chromosome than chromosomes X and 14, based on Bryant and collaborators RS dataset[45] and using BWA aligner[21]. Here, using bowtie2 aligner (which is more robust compared to BWA[46]) and recently published data from Liu and collaborators[47], which includes GSC and SCI datasets, we observed that H3K9me3 enrichment as spermatocytes specialize into spermatids is higher on the X than on the Y chromosome, and higher on chromosome 14 than other autosomes (Supplementary Fig. 6d).

Overall, our data show that a high proportion of downregulated genes are enriched in H3K79me2 and are located on the X chromosome, suggesting that DOT1L is required for X gene activation following MSCI, likely through de novo H3K79 dimethylation. These genes, whose expression is promoted by DOT1L (i.e. downregulated genes upon *Dot1l* loss), are located in an H3K27ac-rich environment, devoid of H3K27me3. Finally, our data indicate that the X and Y chromosomes exhibit distinct chromatin remodeling dynamics in male germ cells, with H3K79me2 being one of the most dynamic histone marks involved in the sex chromosome's epigenetic remodeling during spermatogenesis.

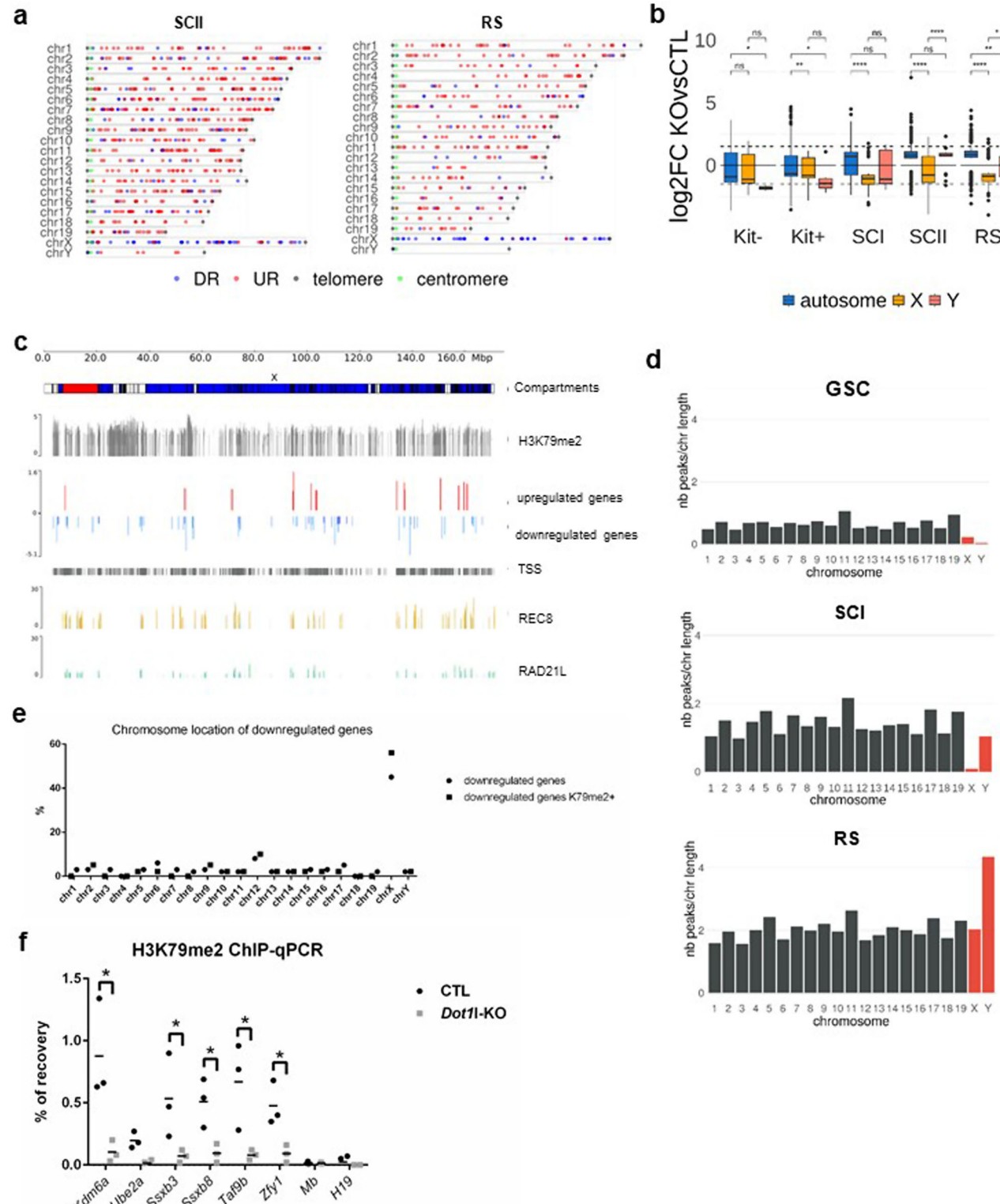

## Discussion

DOT1L is an essential regulator of development and differentiation, but its underlying molecular mechanisms remain unclear. In recent years, several studies have investigated the link between DOT1L, H3K79 methylation, and transcription regulation using transcriptomic analyses[38,48–52]. These studies indicate that DOT1L, initially thought to be a transcription activator, has more diverse effects on gene expression, with common and specific features depending on the studied cell type[6].

In the male germline, DOT1L is involved in several steps: before meiosis, in the self-renewal of spermatogonial stem cells and, after meiosis, in the regulation of histone-to-protamine transition in spermatids and their proper differentiation into functional spermatozoa[18,22,53]. The transcriptional and epigenetic landscapes are highly dynamic during male germ cell differentiation, and DOT1L and H3K79 methylation levels are particularly high. Consequently, spermatogenesis serves as a unique model to investigate the link between DOT1L, H3K79 methylation, and transcription. To this

**Fig. 5 | DOT1L regulates XY gene expression via H3K79me2. a** Chromosome localization of deregulated genes in SCII and RS. Blue: downregulated genes (DR), red: upregulated genes (UR), the green dot indicates the centromere, and the gray dot, indicates the telomere. **b** Comparison of gene expression levels between *Dot1l*-KO and CTL male germ cells. Boxplots showing gene expression log2-fold change of *Dot1l*-KO relative to CTL (all genes with a *p*-value < 0.05). Box: 25th/75th percentiles. Bar in the box: median. Whiskers: 1.5 times the interquartile range from the 25th/75th percentiles. Dashed lines: 1.5-fold change. Stars indicate *p*-values calculated using the Wilcoxon test adjusted with Benjamini–Hochberg correction (\**p* < 0.05, \*\**p* < 0.005, \*\*\*\**p* < 0.00005, ns: not significant). **c** X-chromosome localization of deregulated genes in RS (upregulated are shown in red and downregulated, in blue). Different genomic features are displayed: A/B compartments, H3K79me2 peaks, cohesin peaks (REC8 in yellow and RAD21L in green), and

transcription start sites (TSS). **d** Number of H3K79me2 peaks, normalized to the chromosome length, per chromosome in GSC, SCI, and RS. Bars representing the results for sex chromosomes are colored in red. **e** Chromosome location of downregulated genes in *Dot1l*-KO vs. CTL RS indicated in % (circle). The squares indicate the values obtained for downregulated genes, which are marked by H3K79me2 at their gene body, at the TSS, or at the closest distal regulatory region. **f** Quantification of H3K79me2 level at the gene body of downregulated XY genes by ChIP-qPCR in RS (*n* = 3 replicates for CTL and *Dot1l*-KO samples). *Mb* and *H19* are two not deregulated autosomal genes used as controls. Results are presented as % enrichment values (IP/input recovery %), with the mean indicated by a dash. An asterisk indicates a significant difference as determined by multiple *t*-tests corrected using Holm-Sidak method (*p* < 0.05).

aim, we studied the transcriptional changes induced by *Dot1l*-KO throughout spermatogenesis, profiled H3K79me2 at different cell stages (pre-, meiotic, and post-meiotic cells) using ChIP-seq, and integrated these data with male germ cell chromatin environment inferred from six other histone marks.

## H3K79me2 dynamic correlates with male germ cell genetic program

Our ChIP-seq results show that the number of H3K79me2+ regions increases with male germ cell progression and that many peaks are established after meiosis (i.e. a ~3-fold increase in RS compared to GSC). This pattern correlates with the increase in DOT1L protein level observed at this stage[18–21], indicating that, although DOT1L is presumed to have other functions[6,54], its H3K79 methyltransferase activity is a major consequence of its expression during spermatogenesis.

Our data also show that cell stage-specific H3K79me2 peaks are located in genes with functions consistent with the corresponding cell-specific transcriptional program, such as DNA repair, chromosome segregation, and nuclear division for SCI, and cilium and spermatid development for RS. This demonstrates that the H3K79 methylation genomic landscape is highly remodeled throughout spermatogenesis and that the redistribution and acquisition of H3K79me2 is linked to the ongoing developmental biological process.

It has been widely described that H3K79me2 is enriched at the gene body of transcriptionally active genes, but this pattern is not exclusive, as H3K79 methylation has also been reported at enhancers, at replication origins and at repetitive elements[16,24,50,51]. In male germ cells, we observed the typical H3K79me2 enrichment at gene body (including the promoter), but also at distal intergenic regions. Interestingly, as spermatogenesis advances, these regions reach ~15% of all peaks found at the RS stage and account for ~25% of RS-specific peaks.

Using published ChIP-seq datasets to define male germ cells' chromatin states, and in particular those corresponding to enhancers, we show that all types of H3K79me2+ enhancers are associated with a higher expression of nearby genes compared to H3K79me2− enhancers, in line with Godfrey et al. findings in MLL cell lines[51]. When investigating the link between H3K79me2 and *Dot1l*-KO-induced gene deregulation, we observed that the majority of downregulated genes are enriched in H3K79me2 at their body and/or enhancer, in agreement with the positive correlation between H3K79me2 and gene expression. Upregulated genes, however, were less frequently enriched in H3K79me2, and, when present, H3K79me2 was more often enriched at enhancer regions than at gene bodies, suggesting that the association of H3K79me2 with gene expression depends on its localization and on its chromatin environment (see also below). These results are in agreement with those obtained by Cattaneo et al. on cardiomyocytes, where H3K79me2 at gene bodies and enhancers was associated with gene activation, while H3K79me2 restricted to enhancer regions was associated with gene repression[50].

Overall, our results, along with previous findings, confirm that H3K79 methylation defines cell-specific transcriptional signatures during differentiation processes, including spermatogenesis. ChIP-seq in mouse

spermatozoa is particularly challenging due to the high level of compaction of their chromatin and the limited number of retained histones, but it would be interesting to follow H3K79me2 enrichment in mature spermatozoa and after fecundation, especially since we found that it marks most bivalent regions in RS, regions which were found to be associated with poised developmental genes[55].

## DOT1L repressive effect at enhancers is mediated by an H3K27me3-dependent process

Using a spike-in normalization, we investigated the consequences of *Dot1l*-KO on the transcriptome of five different male germ cell types (i.e. Kit−, Kit +, SCI, SCII, and RS). Strikingly, at all stages, the loss of *Dot1l* led to a higher proportion of upregulated genes than downregulated ones. In somatic cells, similar results were obtained by others[38,52,56–58] but, to the best of our knowledge, our study is the first one that uses spike-in normalization to confirm the transcriptional shift towards upregulation. While this upregulation appears widespread, it is of small intensity. This indicates that DOT1L is unlikely involved in the primary control of gene expression but rather in its modulation.

Our combined analysis of the enrichment in six histone marks assessed by ChIP-seq uncovered distinct epigenetic signatures at the promoters of differentially expressed genes, which we validated by ChIP-qPCR and quantitative CUT&Tag. Downregulated genes are in a local chromatin environment characterized by high levels of H3K79me2, H3K4me3, and H3K27ac, with almost undetectable levels of H3K27me3. Conversely, upregulated genes are enriched in H3K27me3 and particularly present in (poised) bivalent regions, while their local levels in H3K79me2 and H3K27ac are low (see Fig. 6). DOT1L, therefore, appears to cooperate with H3K27me3 to prevent premature gene expression. In this model, at progenitor stages, DOT1L would repress the late germ cells' transcriptional program, and, indeed, our gene set enrichment analysis showed that, in premeiotic cells (i.e. Kit− and Kit+ spermatogonia), deregulated genes are involved in processes associated with the postmeiotic differentiation of spermatids in spermatozoa, such as sperm motility and capacitation, acrosome assembly and reaction, flagellum and cilium organization and assembly, etc. Consistently with the findings of Lin et al.[22], we found only a small number of genes deregulated in *Dot1l*-deficient spermatogonia. However, none of these genes are directly related to defects displayed by spermatogonia.

Our data also suggest a shift from H3K27me3 to H3K27ac at deregulated genes in the case of *Dot1l* KO. Interestingly, we found that the expression of *Hdac8*, an X-linked gene coding for an H3K27 deacetylase, is activated by DOT1L. HDAC8 has been shown to repress the expression of certain genes through its capacity to deacetylate enhancers[39]. In the early embryo, it can prevent the premature expression of developmental genes[40]. DOT1L could, therefore, prevent the expression of hundreds of genes by impeding H3K27 acetylation and promoting H3K27 trimethylation. The fact that the gene coding for the H3K27 demethylase, *Kdm6a*, is, like *Hdac8*, downregulated in KO SCII and RS[18] and the present study, though not confirmed by qRT-PCR (Supplementary Fig. 6c), suggests a complex crosstalk that will require further investigation.

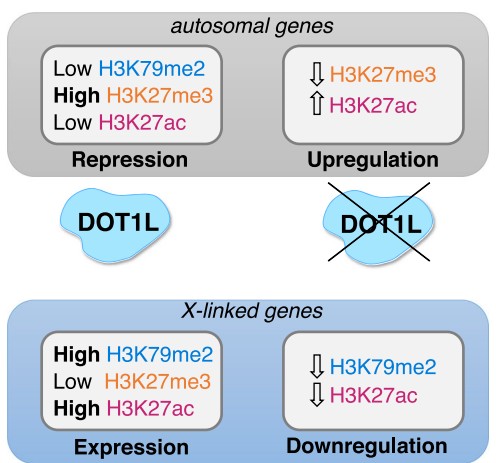

**Fig. 6 | The impact of DOT1L on gene expression depends on the local chromatin environment of these genes.** Model figure summarizing the findings of this article.

Finally, we confirmed the identification of *Bcl6* among DOT1L direct targets[18] and identified other repressors, such as *Bcorl1*, which encodes a BCL6 co-repressor that can interact with HDAC[59], and *Zkscan17* (also known as *Nizp1*), a gene which is highly expressed from the meiotic stage, and encodes a KRAB domain zinc finger protein of yet unknown role during spermatogenesis. This result indicates that DOT1L repressive effect is likely mediated by several pathways, in agreement with the repressive function of DOT1L observed by others[54].

### DOT1L activates X chromosome gene expression by an H3K79me2-dependent mechanism

A specific epigenetic regulation of the sex chromosomes takes place during mammalian spermatogenesis, where X and Y chromosomes are transcriptionally silenced during meiosis, known as the meiotic sex chromosome inactivation (MSCI)[41]. After meiosis, a number of sex chromosome genes are reactivated or expressed de novo[43].

Our study provides novel insights into the mechanism of XY gene regulation during spermatogenesis. First, we identify H3K79me2 as a hallmark of the post-MSCI state, as it is highly enriched in sex chromosomes compared to autosomes at the postmeiotic stage, more than any other studied histone mark (H3K27ac, H3K27me3, H3K4me1, H3K4me3, or H3K9me3). Furthermore, we show that DOT1L promotes the expression of genes predominantly encoded on the X chromosome and enriched in H3K79me2, suggesting that H3K79me2 contributes to the post-MSCI activation of X-linked genes. These results align with the pattern of DOT1L expression, which has been shown to co-localize with the sex chromosomes towards the end of MSCI (from the diplotene stage onwards) in meiotic and postmeiotic cells[21,35].

Surprisingly, the Y chromosome seems to be regulated differently, with very few Y genes significantly deregulated in *Dot1l*-KO male germ cells. Additionally, the Y chromosome-encoded multicopy gene *Sly* is repressed by DOT1L in SCII and RS. *Sly* encodes a repressor of sex chromosome gene expression[60]; its upregulation in *Dot1l*-KO cells could, therefore, contribute to the downregulation of X genes and of a few Y genes such as *Zfy1*.

In summary, we demonstrate that DOT1L has opposite effects on gene expression depending on the gene's local chromatin environment. The differentiation of male germ cells is well-suited to unravel this phenomenon, as autosomes and sex chromosomes have very distinct and dynamic chromatin marks during this process[29,43]. On one hand, DOT1L is involved in the repression of autosomal genes associated with H3K27me3 in bivalent regions. On the other hand, it contributes to the activation of X-linked genes following MSCI, concomitant with the acquisition of H3K79me2, H3K27ac, and H3K4me3. The genes regulated by DOT1L include genes encoding for well-known transcriptional repressors, such as BCORL1 and ZKSCAN17, and chromatin regulators, such as HDAC8, as well as for SLY, which itself

represses the postmeiotic expression of XY genes. Altogether, these data indicate that the impact of DOT1L on gene expression is complex and multi-layered.

## Methods

### Sample collection

Cells were isolated from Dot1l conditional knock-out and control C57/Bl6 male mice. We have complied with all relevant ethical regulations for animal use. Authorization was obtained from the ethical committee of Universite Paris Cité (Comite d'Ethique pour l'Experimentation Animale; registration number CEEA34.JC.114.12, APAFIS 14214-2017072510448522v26)[18].

**For ChIP-seq.** H3K79me2 ChIP-seq analyses were performed on male germ cells of four different types: undifferentiated spermatogonia grown in culture (GSC), primary spermatocytes (SCI), round spermatids (RS) and elongating/condensed spermatids (ES) (see Fig. 1a). Cells were isolated from adult (3–5-month-old) wild-type C57/Bl6 male mice by elutriation, as described in Cocquet et al.[60] In brief, testes from three adult mice per biological replicate were dissected and chopped in DMEM (GIBCO) then treated with 2.5 mg/ml trypsin (GIBCO) and 50 mg/ml DNase I (Sigma) for 30 min at 31 °C with stirring on a ThermoMix at ~300 rpm. Fetal calf serum (GIBCO) was then added (8% finale), and the cells were passed through a 40-µm nylon filter (Millipore). Cells were then pelleted and resuspended in DMEM 0.5% bovine serum albumin (Sigma) with 50 mg/ml DNase I, and cooled on ice. Enriched fractions in SCI (~70% purity), RS (~80–90% purity), and ES (~95–99% purity) were obtained using a standard chamber with a JE-5.0 rotor in a J-6M/E centrifuge (Beckman). ES were collected at 3000 rpm with a flow rate of 16 ml/min, RS at 3000 rpm with a flow rate of 40 ml/min and SCI at 2000 rpm with a flow rate of 28–30 ml/min. Collected fractions were washed in 1× PBS and then used for subsequent ChIP-seq analyses. To derive the primary culture of adult GSC, α-6+Kit−β2M− germinal cells were isolated from 3 mice by flow cytometry and maintained on mitomycin-C-treated MEFs at 37 °C and 5% CO$_2$ as previously described[61]. The GSC culture medium was composed of Stem Span (Stemcell Technologies) and B27 supplement (Life Technologies), supplemented with recombinant human GDNF (40 ng/ml, R&D Systems), recombinant rat GFRα1 (300 ng/ml, R&D Systems), FGF2 (1 ng/ml, Life Technologies), and ES-Cult™ fetal bovine serum (1%, Stemcell Technologies). GSCs were used for experiments after at least 1.5 months of culture. H3K4me1 ChIP-seq experiments were performed on undifferentiated Kit− and differentiating Kit+ spermatogonia, which were flow sorted according to the expression of α-6 integrin, the receptor Kit and β2-microglobulin as described in Barroca et al.[61] and in the next paragraph.

**For CUT&Tag and RNA-seq.** CUT&Tag and RNA-seq analyses were performed on male germ cells from *Dot1l* conditional knock-out (KO) and control mice (CTL) as described in Blanco et al.[18]. For RNA-seq, male germ cells of five different types [undifferentiated spermatogonia (Kit−), differentiated spermatogonia (Kit+), primary spermatocytes (SCI), secondary spermatocytes (SCII) and round spermatids (RS)] were isolated from adult (3–5-month-old) males by flow cytometry sorting according to their size, DNA content, and the phenotypes α-6+Kit−β2M− and α-6+Kit+β2M− as described in Barroca et al.[61] and Bastos et al.[62]. Testicular cell suspensions from adult mice were prepared according to previous protocols[62–64]. Briefly, the seminiferous tubules were gently isolated using enzymatic digestion by collagenase type I (100 U/ml, 25 min at 32 °C) in Hanks' balanced salt solution (HBSS) supplemented with 20 mM HEPES pH 7.2, 1.2 mM MgSO$_4$·7H$_2$O, 1.3 mM CaCl$_2$·2H$_2$O, 6.6 mM sodium pyruvate, 0.05% lactate. To discard the interstitial somatic cells, the seminiferous tubules were collected using a 40 µm filtration step. The tubules were further incubated in a cell dissociation buffer (In Vitrogen) for 25 min at 32 °C. The whole cell suspension was filtered through a 20 µm filter. After an HBSS wash, the cell pellet was

resuspended in incubation buffer (HBSS supplemented with 20 mM HEPES pH 7.2, 1.2 mM $MgSO_4·7H_2O$, 1.3 mM $CaCl_2·2H_2O$, 6.6 mM sodium pyruvate, 0.05% lactate, glutamine, and 1% fetal calf serum) and further incubated at 32 °C in a water bath.

For the isolation of germinal cells, testicular cells expressing α-6 integrin were first pre-enriched using immunomagnetic selection. This involved labeling the cells with anti-α-6 integrin-PE (GoH3) (BD Pharmingen) and anti-PE microbeads (Miltenyi Biotec), following the manufacturer's instructions. To sort spermatids, spermatocytes I, and spermatocytes II based on their DNA content, Hoechst staining (5 µg/ml) was applied to the α-6- cell fraction, as described in previous studies[62–64]. Cells in the α-6+ fraction were labeled with β2m-FITC (Santa Cruz) to exclude somatic cells, and anti-CD117-APC (2B8) (BD Pharmingen) was used to distinguish undifferentiated (α-6+c-Kit−/lowβ2m−) from differentiating (α-6+c-Kit+β2m−) spermatogonia. Propidium iodide (Sigma) was added prior to cell sorting to exclude dead cells. Cell sorting was performed on ARIA II flow cytometers (Becton Dickinson). For CUT&Tag experiments, RS were collected using the same procedure.

## ChIP-seq
Cells were first fixed with 1% formaldehyde for 10 min then quenched with 0.125 M glycine. Pellets were then washed in 1× PBS before being snapped-frozen in liquid nitrogen and transferred to −80 °C for storage. Fixed cells were processed using Diagenode iDeal ChIP-seq kit for SCI, RS, and ES cells, or True MicroChIP kit for GSC, Kit− and Kit+ cells, according to the manufacturer's instructions. ChIP assays were performed using 0.5 µg of H3K79me2 antibody (Diagenode C15410051, Lot A1193) per 1 million cells. For Kit− and Kit+ cells, ChIP assays were performed using 100 ng of chromatin per IP with 0.5 µg of H3K4me1 antibody (Diagenode C15410194, Lot A1862D). For GSC, ChIP assays were performed using 500 ng of chromatin per IP with 1–2 µg of H3K79me2 antibody (Diagenode C15410051, Lot A1193). In total, four replicates for GSC and duplicates for SCI, RS, and ES were produced on H3K79me2. Duplicates for Kit− and Kit+ flow-sorted cells isolated from 5 mice per replicate were also produced on H3K4me1 and one input sample was generated by cell type. Libraries were prepared following the MicroPLEX (Diagenode) protocol instructions and sequenced on an Illumina Hiseq 4000 (50 bp, single end, >25 million reads per sample).

## RNA-seq
Total RNA was extracted using the RNAqueous kit (Thermo Fisher Scientific) following the manufacturer's protocol. In total, three replicates were generated for Kit− and Kit+ cells, and five replicates for SCI, SCII, and RS. For "spike-in" analyses, external RNA controls consortium (ERCC) was added to RNA samples according to the number of cells used for RNA isolation, and correcting for the ploidy, following the manufacturer's instructions (ERCC RNA Spike-In Mix, Thermo Fischer Scientific). ERCC are exogenous synthetic polyadenylated transcripts of 25–2000nts which can be added to samples before library preparation to serve as internal standards for RNA-seq. The transcripts are designed to mimic natural eukaryotic mRNAs. ERCC was added in the three replicates of each cell type. Libraries were prepared for all samples using the NEB Next Ultra II Directional RNA Library Prep Kit (New England Biolabs) according to supplier recommendations. Paired-end sequencing of 100-bp reads was then carried out on the Illumina HiSeq4000 system.

## RT-qPCR
RT-qPCR was performed on RNA extracted from Dot1l-KO and CTL round spermatids and reversed-transcribed using SuperScript® IV Reverse Transcriptase as described by the manufacturer (Thermo Fischer Scientific). Quantitative PCR was performed on a Roche LightCycler 480 using SensiFast No-Rox kit mix (Bioline) following instructions from the manufacturer. Values were normalized to the geometric mean of β-actin and Ubb. Primer sequences and qPCR conditions for Slx, Slxl1, Ssty1, Ssty2, Ubb are from Ellis et al.[65], β-actin from Garcia et al.[66], Acrv1 from Akerfelt et al.[67],

Zfy1 from Vernet et al.[68], Zfy2 from Vernet et al.[69], Hdac8 from Pantelaiou-Prokaki et al.[70] and Kdm6a from Berletch et al.[71]. Other primer sequences and qPCR conditions can be found in Supplementary Data 5.

## Western blot
For protein detection, proteins extracted from purified germ cells were denatured 5 min at 95 °C in 4× NuPage LDS sample buffer (Thermo Fischer scientific) supplemented with 10% β-mercaptoethanol. Extracts were then loaded on SDS–polyacrylamide gels for electrophoresis (SDS–PAGE) and subsequently transferred to a nitrocellulose membrane. Blocking was performed incubating 1 h at room temperature in 5% skimmed milk in 1× PBS containing 0.01% Tween. The membranes were cut in halves and incubated at 4 °C overnight with anti-tubulin antibody (05-661, Upstate, diluted 1/3000) for the upper part, and with anti-H3K79me2 antibody (ab3594 Abcam diluted 1/500) for the bottom part. Secondary antibodies (goat anti-mouse HRP or goat anti-rabbit HRP, references 31460 or 31430, Thermo Fischer Scientific) were incubated for 2 h at room temperature. Imaging was performed with Immobilon ECL Ultra Western HRP substrate (Millipore) on ImageQuant™ LAS 4000 imager (GE Healthcare) or exposing the membrane to film. The western blot images shown in Supplementary Fig. 4 are unmodified. Uncropped images are shown in the source data file.

## Spike-in CUT&Tag
Fresh FACS-sorted round spermatids from Dot1l-KO and control mice were used in triplicates. A negative control (using rabbit anti-IgG, 02-6102, Thermo Fischer Scientific) was performed using a mix of round spermatids from the three Dot1l-KO biological replicates. After sorting, cells were centrifuged 500×g 10 min at 4 °C and resuspended in 20 mM HEPES pH 7.5, 150 mM NaCl, 0.5 mM Spermidine, 1 mM sodium butyrate, and 1X Protease inhibitor cocktail (Roche) for counting. CUT&Tag-IT® Spike-In Control (53168, Active Motif), consisting of the use of Drosophila nuclei for downstream normalization for quantitative comparison, was added in a 1:20 spike-in:cells ratio to each sample. The mix of mouse round spermatids and Drosophila nuclei was subjected to CUT&Tag as described by Kaya-Okur and colleagues[72] with minor modifications. BioMag®Plus Concanavalin A-coated magnetic beads (86057-3, Polysciences), previously activated with 20 mM HEPES pH 7.9, 10 mM KCl, 1 mM $CaCl_2$, 1 mM $MnCl_2$, were mixed with the cells and incubated during 10 min at room temperature (RT) with constant rotation. Supernatant was discarded and bead-bound cells were resuspended in 20 mM HEPES pH 7.5, 150 mM NaCl, 0.5 mM Spermidine, 0.05% Digitonin, 2 mM EDTA, 0.1% bovine serum albumin, 1 mM sodium butyrate, and 1X Protease inhibitor cocktail. The antibody of interest (Abcam rabbit anti-H3K27ac ab4729, Cell Signalling Technologies rabbit anti-H3K27me3 9733, or Millipore rabbit anti-IgG CS200581 for negative control) and the anti-H2Av Spike-in antibody specific to Drosophila nuclei were added in 1:50 dilution to each sample, and incubated overnight at 4 °C under constant rotation. Primary antibody solution was removed, and cells were incubated for 30 min at RT on a nutator mixer with Guinea Pig anti-Rabbit IgG (H + L) Secondary Antibody (ABIN101961, Antibodies Online) at 1:100 dilution in dig-wash buffer (20 mM HEPES pH 7.5, 150 mM NaCl, 0.5 mM Spermidine, 0.05% Digitonin, 1 mM sodium butyrate, 1X Protease inhibitor cocktail). Cells were washed three times with dig-wash buffer before incubation with loaded pAG-Tn5 Transposase (C01070001-30, Diagenode) diluted 1:250 in dig-300 buffer (20 mM HEPES pH 7.5, 300 mM NaCl, 0.5 mM Spermidine, 0.05% Digitonin) for 1 h at RT on a nutator. Cells were washed three times with dig-300 buffer and resuspended in tagmentation buffer (10 mM $MgCl_2$ in dig-300 buffer) to activate tagmentation reaction, incubating 1 h at 37 °C in a thermoblock. Right after, 10 µL 0.5 M EDTA, 3 µL 10% SDS, and 2.5 µL 20 mg/mL Proteinase K were added to each sample to stop the reaction, incubating 1 h 55°C in a thermoblock. DNA was extracted using ZYMO Research kit DNA Clean & Concentrator (#D4014, ZYMO Research) and eluted in 25 µL 1 mM Tris–HCl pH 8, 0.1 mM EDTA. Libraries were prepared by mixing 25 µL of NEBNext High-Fidelity 2x PCR Master mix (M0541, New England Biolabs), 21 µL CUT&Tag DNA, and 2 µL of each

universal i5 and uniquely barcoded i7 primers from Buenrostro et al.[73]. Samples were placed in a thermocycler with heated lid, using the following cycling conditions: 72 °C 5 min, 98 °C 30 s, 13 cycles of 98 °C 10 s and 63 °C 30 s, the final extension of 72 °C 1 min, and hold at 8 °C. Post-PCR size-selection and clean-up was performed by adding 0.9x AMPure XP beads (A63880, Beckman Coulter) to the amplified libraries, incubating 10 min at RT, washing twice with 80% ethanol, and eluting in 25 μl of 1 mM Tris–HCl pH 8, 0.1 mM EDTA. Libraries were quantified by Qubit (Thermo Fisher Scientific) and mixed to achieve equal representation. Library distribution was determined using a 2100 Bioanalyzer system (Agilent) before performing paired-end (2 × 50 bp) sequencing on Illumina NextSeq 2000 P2 with 1% Phi-X at the genomic facility of Institut Cochin (Genom'IC).

## ChIP-qPCR

Chromatin immunoprecipitation was performed on round spermatids isolated from adult (3–5-month-old) *Dot1l*-KO and control mice by elutriation, as described in Cocquet et al.[60] and above. Each replicate represents a pool of 2–3 mice. Cell purity was checked under the microscope, and elutriated fractions containing above 70% round spermatids and <11% primary spermatocytes were used. ChIP against H3K79me2 was run in triplicates, using 1% formaldehyde-fixed flash-frozen cells. ChIP was performed using iDeal ChIP-seq kit for transcription factors (Diagenode C01010055), and resulting DNA was purified using IPure Kit v2 (Diagenode C03010015) following the manufacturer's recommendations. Anti-H3K79me2 antibody (Diagenode C15410051) was used in a ratio of 1 μg antibody per 5M cells.

Real-time quantitative PCR was performed using a SensiFAST SYBR No-ROX mix 2X (BIO-98005, Meridian Bioscience) in a Roche LightCycler 480 (Roche). ChIP and input DNA were diluted so that input represented 1% of ChIP material, hence the enrichment was calculated as $2(Ct_{input} - Ct_{sample})$. Primers were designed to amplify regions across the TSS of indicated genes. Primer sequences and qPCR conditions for *Ube2a* are from Murphy et al.[74]. Other primer sequences and qPCR conditions are shown in Supplementary Data 5.

## ChIP-seq analyses

Sequences adaptors were trimmed and read with poor quality (quality < 20) and were filtered out with BBduk from BBTools (v. 38.23, https://sourceforge.net/projects/bbmap/). Alignment was performed on the mouse reference genome (mm10) using bowtie2[75] with the following arguments: --local. Peak calling was processed against the input file using the MACS2 tool[76] with a "broad pattern" setting to obtain a peak enrichment score of H3K79me2. The parameter "broad" was also used for H3K36me3 and H3K27me3 histone marks. Mitochondrial, ambiguous and peaks with FDR > 5% were removed. Peaks were annotated with the R package ChIPseeker[77] using the annotatePeak function with the following parameters: tssRegion = $c(-3000,3000)$, and divided into four categories: promoter (promoter's annotations), intragenic (exon, intro and UTR), downstream and distal intergenic. Snakemake[78] (v. 3.9.0) was used to run this ChIP-seq pipeline. Common peaks between replicates were defined using BEDtools intersect function[79] (v. 2.29.2).

To study dynamics, regions with a minimum overlap of 1 bp between two cell types (using BEDtools merge function) were considered in common. The size of these new regions was similar to the size of H3K79me2 peaks (Supplementary Fig. 7a). Ontology analyses for H3K79me2 genic peaks were performed with compareCluster function from clusterProfiler package[80] using BP (biological processes) database and Benjamini–Hochberg correction to calculate the adjusted *p*-value.

## Spike-in CUT&Tag analyses

Data were analyzed using the same alignment pipeline described for ChIP-seq data on both the mouse genome (GRCm38.p6) and the Drosophila genome (BDGP6.46). H3K27ac and H3K27me3 peaks enrichment were normalized using the Drosophila enrichment score. To establish the enrichment profile of H3K27ac according to the different types of

enhancers, we used enhancers as defined by the ChromHMM tool (see below). For each category of enhancer with and without H3K79me2, the level of H3K27ac enrichment for the different samples was represented using the Deeptools package (v. 3.3.2)[81].

## Characterization of chromatin states using hidden Markov modeling

ChromHMM[36] was used with default parameters to predict the chromatin state associated with each genomic region (parameter bin size = 200 bp). We created a model of 18 chromatin states by integrating the data from six histone marks (H3K4me1, H3K4me3, H3K27ac, H3K27me3, H3K36me3, and H3K9me3) originating from 17 different cell types/stages, retrieved from the literature and the mouse ENCODE project (Consortium, 2012). ChIP-seq data from male germ cells were obtained from refs. [47,82–84]. Chromatin states were named according to ChromHMM nomenclature[36,51,57,85]. Four chromatin states were defined as enhancers: active enhancers (EnhA), genic enhancers (EnhG), weak enhancers (EnhW), and bivalent enhancers (EnhBiv). Each type was subdivided into H3K79me2+ or H3K79me2−, according to the enrichment score from H3K79me2 ChIP-seq data. To assign genes to each category of enhancers, two methods were used: the first one, based on Rada-Iglesias et al.[86], assigns each enhancer to the nearest gene within 100 kb from each side of the enhancer boundaries. The second one associates each enhancer with all genes found in the same interval, considering that one enhancer may regulate the expression of several genes[87]. We used the ChromENVEE package (https://github.com/ManonCoulee/ChromENVEE) to calculate the average expression level (CPM) of genes within the interval of ±100 kb from the enhancer of interest with enhancerExpression function. Statistical analysis is performed using the Wilcoxon test between enhancers with (+H3K79me2) and without H3K79me2 (−H3K79me2) (Supplementary Fig. 3). geneEnvironment function with default parameters was used to identify the chromatin states at promoter regions of genes deregulated in *Dot1l* conditional knock-out (KO) vs. control (CTL) samples (see below for detailed RNA-seq analyses). Chromatin state was assigned to each deregulated gene by determining the chromatin environment around the TSS (±3 kb) and selecting as the predominant state the one with the higher percentage of coverage. The percentage of coverage for the second most predominant chromatin state is drastically inferior to that of the first most predominant state indicating that ambiguous chromatin states are very rare (Supplementary Fig. 8).

## Dynamics of chromatin states during spermatogenesis

Each H3K79me2 region was associated with the chromatin state (defined by ChromHMM tool) with the highest coverage using BEDtools intersect. To study their dynamics, we first reduced the complexity of the alluvial plot, by only keeping dynamic groups containing more than a hundred of peaks. To produce the supplemental figures (Supplementary Fig. 2), categories of chromatin states were defined as follows: actives (TSSA, Tx, EnhG, and EnhA), weaks (TSSFlnk, TxWk and EnhW), heterochromatins (ZNF/Rpts and Het), bivalents (EnhBiv and TssBiv), polycomb (ReprPC and ReprPCWk) and quiescent.

## RNA-seq analyses

Five samples of each genotype (KO and CTL) for SCI, SCII, and RS, and three samples for Kit− and Kit+ were analyzed. Sequence adaptors were trimmed and reads of low quality were filtered out using the same parameters as for ChIP-seq analyses. Alignment was performed on the mouse reference genome build mm10 (GRCm38.p6) using STAR[88] (v. 2.7.2d) and Gencode vM19 gene annotation GTF with the following arguments: --outMultimapperOrder Random --quantMode Genecounts --sjdbOverhang 74. For each sample, the number of aligned reads was converted to cpm (counts per million). To exclude genes with a very low expression level, only genes with an expression level of at least 1 cpm in a minimum of two samples were included in the analysis. Differential expression analysis was carried out using DESeq2[89] and edgeR[90] packages with default parameters

and a design accounting for genotype and cell type-induced variability. Differentially expressed genes (FDR < 5%) were obtained using glmFit and glmTreat edgeR's function, which performs a modified likelihood ratio test (LRT) against the fold change threshold (>1.5) based on the REAT method (McCarthy & Smyth[91]). Deregulated genes with FDR < 5% (adjusted p-value) and a Fold Change >1.5 were selected. Enrichment analysis was performed with GSEA[92] (v. 4.0.3, c5.all.v7.1.symbols.gmt, GO cellular components, biological pathways, or all) by processing gene expression cpm values from KO and control samples using the same parameters as in Blanco et al.[18]. Beforehand, mouse Ensembl gene IDs were converted to human Ensembl gene IDs (hg38) with the bioMart R package[93]. Snakemake[78] (v. 3.9.0) was used to run this RNA-seq analysis pipeline.

### Spike-in RNA-seq analyses
Three samples of each genotype (KO and CTL) were analyzed for Kit−, Kit +, SCI, SCII, and RS (in which ERCC was added prior to library preparation). RNA "spike-in" data were analyzed using the same alignment pipeline described for classic RNA-seq data. Normalization was realized using the RUVg method from RUVseq R package[37] to remove unwanted variation, followed by differential gene expression analysis with the same parameters as described above. It consists of using ERCC values as a control to normalize the data. Differential expression analysis was then performed using the same parameters as described above.

### Gene length, functional annotation
Gene length and functional annotation were performed by comparing deregulated genes with GTF information from mm10 (gencode vM19 gene annotation). For gene length information, we divided gene length into ten categories according to percentile value. Functional annotations were divided into six groups of annotations: IG-TR, Protein coding, Pseudogene, lncRNA, mtRNA and snoRNA. Statistical analyses were performed using the Chi[2] test by comparison of deregulated genes with non-deregulated genes.

### Identification of repression regulator
To identify transcriptional repressor in downregulated genes, we have based on Kwesi-Maliepaard paper[52]. List of downregulated genes enriched in H3K79me2 is compared to one of the following pathways: "negative regulation", "negative regulation of transcription by RNA polymerase II", and "repressor" (pathways defined in Amigo database)[94].

### Multi-association analyses
The statistical association between different genomic features was tested using the regioneR version 1.26[95] and regioneReloaded version 1.2.0[96]R packages as previously described[97]. The permutation tests are based on 10,000 permutations, normalized z-score of −0,05 and p-value < 5%.

Structural genomic data included in the permutation analysis was extracted from public repositories. A/B compartments, TAD boundaries, CTCF and cohesin ChIP-seq datasets were obtained from the NCBI GEO repository, accession number GSE132054 (Spermatocytes I Hi-C: GSM3840082 and Spermatids Hi-C: GSM3840083; Spermatocytes I CTCF, RAD21L, REC8: GSM3840086, GSM3840087, GSM3840088; Spermatids CTCF, RAD21L, REC8: GSM3840089, GSM3840090, GSM3840091). Spermatids DSBs raw data is available at PRJEB20038. ATAC-seq data from Spermatocytes I was retrieved from GSM2751129 and GSM2751130. ATAC-seq data from round spermatids was retrieved from GSM2751133 and GSM2751134. PRDM9 and DMC1 ChIP-seq data were obtained from GSE93955. CpGIsland and TSS positions were extracted from the mouse mm10 RefSeq genome repository.

### Statistics and reproducibility
Statistical tests used for individual experiments are described in the corresponding figure legends and Results section. In brief, all statistical analyses were run and plotted with R (v. 4.0.1) using the following packages: ggplot2 for plotting and stats for statistical tests, such as the Wilcoxon test or Chi[2] test. When necessary, the p-value was adjusted using the

Benjamini–Hochberg correction. Spearman correlations were produced using deeptools package (v. 3.3.2) with the plotCorrelation function.

H3K79me2 ChIP-seq analyses were performed on biological duplicates for SCI, RS, and ES (from wild-type males). H3K79me2 ChIP-seq was performed on four biological replicates for GSC (duplicates using 1 µg of H3K79me2 antibody, and duplicates using 2 µg of H3K79me2 antibody, which were then considered as four replicates since their correlation was high). H3K4me1 ChIP-seq analyses were performed on biological duplicates for Kit− and Kit+ cells (from wild-type males). H3K79me2 ChIP-qPCR experiments were performed on biological triplicates of each genotype (Dot1l-KO and CTL). H3K27ac and H3K27me3 spike-in CUT&Tag analyses were performed on biological triplicates of each genotype (KO and CTL). Spike-in RNA-seq was performed on biological triplicates for all. RT-qPCR experiments were performed on 3 and 6 biological replicates for CTL and KO, respectively. For all analyses, each biological replicate represents a pool of cells sorted from 3 to 5 males of the same genotype.

### Reporting summary
Further information on research design is available in the Nature Portfolio Reporting Summary linked to this article.

## Data availability
RNA-Seq data have been submitted to the ENA repository under the project numbers PRJEB50887 and PRJEB64263 (https://www.ebi.ac.uk/ena/). H3K79me2 ChIP-Seq project number is PRJNA643726 for RS and PRJEB64263 for GSC, SCI and ES. H3K27me3 and H3K27ac CUT&Tag data have been submitted under PRJEB64263 project number. The ChIP-seq data used for ChromHMM are available at PRJNA548107 and PRJNA566429 for spermatogonia, SRP028576, PRJNA428362 and PRJNA566429 for spermatocytes and round spermatids, and at the ENCODE project database for other cell types: ENCSR835OVT, ENCSR349UOB, ENCSR228WLP, ENCSR714JXE, ENCSR050BJO, ENCSR693OUQ, ENCSR189YPN, ENCSR192PSV, ENCSR881OKK, ENCSR199TFQ, ENCSR893YWL, ENCSR273IZZ, ENCSR379NJH (https://www.encodeproject.org/). The Hi-C and RNA-seq and cohesion ChIP-seq datasets were retrieved from previous studies[34], accessible at the NCBI GEO repository: GSE132054 (Spermatogonia Hi-C: GSM3840080; Spermatocytes I Hi-C: GSM3840082 and Spermatids Hi-C: GSM3840083; Spermatocytes I CTCF, RAD21L, REC8: GSM3840086, GSM3840087, GSM3840088 and Spermatids CTCF, RAD21L, REC8: GSM3840089, GSM3840090, GSM3840091).

## Code availability
All scripts used in this paper can be found in the GitHub repository (https://github.com/ManonCoulee/H3K79me2_Coulee_2023).

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

## Acknowledgements

We thank Juliette Hamroune from the Genom'IC facility of Institut Cochin (INSERM U1016, CNRS UMR8104, Université Paris Cité). This work is supported by the Agence Nationale de la Recherche (ANR-17-CE12-0004-01 and ANR-21-CE44-0035 to J.C.), the Fondation pour la Recherche Médicale (SPF201909009274 to C.G. and SPF202309017479 to A.I.), the LABEX Who am I (to M.C., ANR-11-LABX-0071 and Idex ANR-18-IDEX-0001), and the Institut Cochin (PIC to L.E.K). J.C, A.I., and A.R.-H. belong to the COST Action (CA20119) Andronet, which is funded by the European Cooperation in Science and Technology (www.cost.eu). A.R.-H. is funded by the Spanish Ministry of Science and Innovation (PID2020-112557GB-I00), the Spanish Ministry of Economy and Competitiveness (CGL2017-83802-P), the Agència de Gestió d'Ajuts Universitaris i de Recerca (AGAUR, 2021SGR00122) and the Catalan Institution for Research and Advanced Studies (ICREA). L.A.-G. is supported by an FPI predoctoral fellowship from the Spanish Ministry of Economy and Competitiveness (PRE-2018-083257).

## Author contributions

M.C. performed bioinformatics analysis. A.I. and C.G. performed ChIP and Cut&Tag experiments. M.B. performed immunoblots. A.I., C.G., M.B., C.L., C.I.-R., J.C, and L.E.K. purified male cells. L.A.-G. and A.R.-H. performed multi-association analyses. G.M. provided bioinformatic guidance and resources. J.C. and L.E.K. conceived, designed, and coordinated the project. M.C., J.C., and L.E.K. wrote the paper. A.I., A.-H., and P.F. contributed to editing the manuscript.

## Competing interests

The authors declare no competing interests.
