## [Transparent Peer Review file · Communications Biology]

Chromatin Environment-Dependent Effects of DOT1L on Gene Expression in Male Germ Cells

Corresponding Author: Dr Laïla El Khattabi

Version 0:

Reviewer comments:

Reviewer #1

(Remarks to the Author)

Coulee et al investigated the distribution of H3K79me2 across different steps of male germ cell development and the effect of DOT1L knockout on gene expression at these different developmental steps. They also integrate their data with publicly available data of other histone modifications and chromosome conformation. The authors found changes in H3K79me2, both on protein-coding genes and enhancers, during the male germ cell development. DOT1L knockout results in a limited but increasing number of genes being deregulated over the developmental steps, with a trend towards upregulation. Finally, the authors found a difference between how autosomal and sex chromosomes respond to DOT1L loss, with the latter associated with downregulation of genes.

The study is of interest but remains generally descriptive and several of the conclusions made by the authors are, in my opinion, not supported by what they are currently showing in the manuscript.

Comments:

1. I think the statements on the role of DOT1L in inhibiting gene expression should be tone down. Knockout models are notoriously known to be associated with compensatory effects that make it difficult to conclude on the direct functions of a protein. To prove an inhibitory function of DOT1L would require at least experiments to show that DOT1L is binding to these repressed genes and that a rapid degradation/inhibition results in the direct transcriptional activation of these genes.
2. From the authors RNA-seq data, are the upregulated genes found in DOT1L-KO condition activated (no expression in control and expression in DOT1L-KO), upregulated (expressed to some extent in control but more expressed in DOT1L-KO), or a mix of the two? It would have been interesting to know whether these upregulated were also associated with paused RNA polymerase (RNAP) II or are totally devoid of RNAPII but it does not seem that data are available to conclude about this.
3. Could the increase in gene deregulation over the male germ cell development be explained by the deregulation of a few key genes at the start of the male germ cell development? Any defect from the beginning could continue to expand independently of any effects associated with the loss of DOT1L.
4. From Blanco et al, 2023 (previous paper from the research group), H3K79me2 has disappeared at the RS developmental step but there is some low level left at the previous SC step. Do the authors know how much H3K79me2 is left in GSG? More than in SC? What would be their expectation for the levels of H3K79me1 and H3K79me3?
5. As the DOT1L KO condition should result in the loss of most, if not all, H3K79 methylation, the fact that only up to ~1,000 genes are affected at most in one condition more than 1.5-fold seems rather to indicate that H3K79 methylation is generally not really required for controlling gene expression.
6. It is the same issue with the X chromosome, yes most of the downregulated are on the X chromosome but compared to all the genes expressed on the X chromosome, what is the proportion downregulated in the DOT1L KO condition? It looks like at most a few percent of the genes so I am not really convinced DOT1L plays as written in the Discussion a pivotal role in XY gene activation (with point 6 below, it looks like more X chromosome only rather than XY).
7. Rather than sex chromosomes, Figure 5 seems to show that it is the X chromosome only that is associated with downregulated genes in the DOT1L-KO condition.
8. Why did the authors choose H3K79me2 rather than H3K79me1 or H3K79me3 that are also dependent on DOT1L activity? This should be explained in the manuscript.
9. Several histone marks (H3K4me3, H3ac/H4ac, H3K36me3, H3K79me, and H3K9me3/H3K27me3 for low/no expression) can be generally seen as a proxy for RNAPII transcriptional activity. I am wondering if it would not have been more

informative to have performed a RNAPII ChIP-seq in addition/rather than H3K79me2. This would have provided a lot more information on gene expression and on RNAPII initiation/pausing and elongation on protein-coding genes and eRNA activity +/- DOT1L knockout.

Michael Tellier

Reviewer #2

(Remarks to the Author)

Reviewer comments for Coulee et al. 2024:

Manuscript by Coulee et al., entitled "DOT1L has opposite effects on sex chromosome and autosomal gene expression in male germ cells" describes the molecular role of DOT1L in transcriptional regulation of male germ cell development. The manuscript builds on the previous publication (Blanco et al. 2023) and demonstrates in more detail the transcriptional changes associated with DOT1L loss in male germ cells. The major finding of the manuscript is that DOT1L influences the transcriptional outcome by different mechanisms depending on the chromatin environment and chromosomal location. Authors conclude that DOT1L specifically activates genes that are located on sex (XY) chromosomes but represses genes that are present on autosomes. Authors further argue that the regulation on sex chromosomes is directly mediated via H3K79me2, whereas on autosomes DOT1L acts indirectly by influencing the balance between H3K27me3 and H3K27ac chromatin on gene promoters and enhancers. While the gene expression changes and genomic features of H3K79me2 are properly established, some of the results are over concluded and no clear mechanistic insight is provided that can clearly explain differential role of DOT1L on autosomal genes and sex-linked genes. Also, the finding that DOT1L represses some genes and activates others is not novel and has been reported before. Nevertheless, the current manuscript presents a big advancement by including the H3K79me2 ChIP-seq data characterization from different types of germ cells and therefore can be of interest to other researchers in the field. Therefore, for the manuscript to be considered for publication, the following points should be properly addressed.

Major comments:

- 1) Previous literature on role of DOT1L in male germ cell development indicates that DOT1L acts at two different stages in spermatogenesis: 1) at mitotic proliferation of spermatogonial stem cells, and 2) postmeiotic spermatid differentiation (Lin et al., *Genes Dev.*, 2022; Lin et al., *Dev* 2023; Blanco et al., *EMBO Rep*, 2023; and Malla et al., *Dev* 2023). However, the data presented in the current study (Fig 1 and Fig 2c) suggests that majority of H3K79me2 enriched peaks (91%) in GSC are associated with transcriptionally weak or quiescent chromatin (lines 154-157), implying that DOT1L deletion should have a minimal impact on gene expression in (Kit-) undifferentiating spermatogonia (Fig 3b). This does not support the already established role for DOT1L in spermatogonial stem cell (SSC) proliferation and maintenance. Authors should explain why they do not observe gene expression changes specific to SSCs.
- 2) Fig 2d shows that enhancers that are enriched in H3K79me2 promote nearby gene expression compared to H3K79me2 depleted enhancers (lines 177-180). But to conclude that H3K79me2 imparts stronger activity to H3K79me2 enriched enhancers, authors need to show that the enhancer activity is altered in the absence of DOT1L by comparing the enrichment of H3K4me1 and/or H3K27ac on some of these enhancer regions (by ChIP-qPCR).
- 3) In Fig 4b, authors demonstrate that the enhancers of upregulated genes are enriched in H3K79me2 and conclude that the repressive effect of DOT1L on gene expression is independent of H3K79me2 (lines 259 - 262). But this conclusion could be drawn only for genes that are completely devoid of H3K79me2 and not for the genes whose expression is driven by H3K79me2 enriched enhancers. Furthermore, if authors suggest that the gene upregulation effect observed in Dot1LKO germ cells is guided by H3K79me2 enriched enhancers, this could mean that H3K79me2 in the wild type cells is repressing these enhancers. Authors should define role of H3K79me2 at these enhancers and explain how they contribute to gene regulation in Dot1LKO cells.
- 4) Based on Fig 4a, Authors conclude that H3K79me2 is more enriched in genes that are not deregulated than the differentially regulated genes (lines 243-246). This is problematic because it contradicts the existing notion in the field that H3K79me2 is positively associated with transcriptional regulation. Also, the conclusion that most of H3K79me2-enriched genes do not exhibit any change in expression in Dot1L deleted cells could be interpreted to mean that the downregulation following DOT1L loss itself is not specific. Authors should explain how H3K79me2 loss can be selectively imparting gene expression at some genes (downregulated genes) and not others (unaltered genes).
- 5) Fig 4c and 4e. Authors conclude that upregulated genes are enriched in H3K27me3, and downregulated genes are enriched in H3K4me3 states in the wildtype cells (lines 280 – 283 and 443 - 449). This indicates that DOT1L somehow impacts deposition of H3K27me3 on genes that it represses, and that the gene upregulation observed in Dot1LKO cells is due to a shift from H3K27me3 chromatin to H3K27ac chromatin at the deregulated genes (lines 451-452). But no H3K27me3 or H3K27ac enrichment in the Dot1LKO cells is shown. Authors should therefore provide H3K27me3 and H3K27ac enrichment (by ChIP-qPCR) at some of the DEGs in Dot1LKO cells to test if these modifications are differentially regulated in Dot1LKO germ cells. This data is needed to establish the H3K79me independent role for DOT1L at genes that are upregulated upon its loss.

Minor comments:

- 6) There is a typo correction in line 96. it should be as..."iii) round spermatids (RS), and iv) elongating spermatids/condensed spermatids (ES) as postmeiotic..."
- 7) Figures are presented in a nonuniform alphabet case. Some figure panels are presented with upper case letters such as

in Fig. 3 and Fig. 5, and others with lower case such as Fig.1, Fig.2 and Fig. 4. This should be corrected.

8) According to Fig 2B and 2C, Quiescent chromatin states are represented by both H3K79me2 enriched and H3K79me2 devoid regions. Can authors comment on what it means in terms of their gene expression regulation by H3K79me2 or upon Dot1l deletion.

9) In lines 161-166 (page 6), authors mention that 70% of H3K79me2 enriched regions that are bivalent were specifically gained in round spermatids compared to 34% in SCI. What could be the impact of DOT1L loss (or loss of H3K79me2) on these bivalent regions with respect to their gene expression. Based on their RNA Seq data, authors should demonstrate whether there is any change in the expression of poised chromatin in SCI and RS when DOT1L depleted.

10) Authors should provide a detailed protocol for the isolation of different testicular cell types by flow cytometry. Authors should clearly indicate how were secondary spermatocytes (SCII) distinguished from other diploid cells like Sertoli cells and other somatic cells. Also, authors should provide a western blot image for the specific markers of each cell type to confirm the percent purity of separation for each cell type that was isolated by either cell sorting or elutriation.

11) According to Fig 3B, the effect of DOT1L loss upon gene expression is greater in secondary spermatocytes (SCII) (more DEGs) than primary spermatocytes (SCI) or round spermatids (RS). Therefore, relative enrichment of H3K79me2 in SCII should be shown (by immunohistochemistry).

12) Authors mention that most of the genes that are deregulated in Kit+ spermatogonia are not differentially regulated in meiotic or postmeiotic germ cells (lines 222-224). Can authors comment on Why that is the case. Is it because these genes are not expressed in meiotic and postmeiotic stages.

13) In Figs 5a and 5b, authors report that sex-linked genes are over-represented in downregulated gene list in Dot1lKO cells (lines 306-315). However, these sex-linked genes are also significantly down regulated in primary spermatocytes when MSC1 is inactive and sex chromosome transcription is inhibited (Fig. 5b). Ideally there should be no change in sex linked genes in primary spermatocytes (SCI) since these are naturally turned off due to MSC1. Authors should comment on why sex-linked genes are still down in SCI.

14) Highest enrichment of H3K79me2 is detected at Y chromosome yet highest gene down regulation is observed on X chromosome (Fig 5d). According to this data, if sex chromosome gene activity is directly dependent on de novo H3K79me2, as is suggested, then the Y linked genes should exhibit highest down regulation.

15) Fig S8 is provided but not mentioned anywhere in the text. It should be removed or discussed in the text.

16) Authors propose that DOT1L mediates gene repression by limiting H3K27 acetylation (and hence promoting H3K27me3 deposition) via expression of Hdac8 (lines 455-460). Authors should explain how downregulated Hdac8 in Dot1lKO cells is specifically acting on autosomal genes only and not acting on sex linked genes that are downregulated.

17) Authors discuss that Kdm6a is a H3K27 methylase; and that it is downregulated in Dot1lKO Secondary spermatocytes and round spermatids (line 457-460). Firstly, KDM6A is not a H3K27 methylase but a H3K27me3 demethylase. Authors should correct that. Secondly, down regulation of KDM6A should lead to enrichment of H3K27me3 in DOT1LKO cells since it is a H3K27 demethylase. The conclusion that there is a shift from H3K27me3 based chromatin to H3K27ac chromatin in Dot1lKO cells does not make sense in the context of the proposed model.

Version 1:

Reviewer comments:

Reviewer #1

(Remarks to the Author)

The authors have answered all my comments.

Reviewer #2

(Remarks to the Author)

Authors have done an excellent job of properly addressing all my previous comments. The revised manuscript includes additional data sets and explanations that clearly validate the findings presented and greatly improve the quality and scope of the manuscript. The updated manuscript could be good resource guide for other researchers working in the field. I therefore recommend the revised manuscript for publication in Communications Biology. Authors can note the following clarification with respect to my previous comment.

In response to my previous major comment 1, authors rightfully argue that the smaller number of differentially expressed genes observed in GSC agrees with previously reported literature (Lin et al. 2022), and that these fractional changes does not preclude the GSCs from exhibiting the phenotype. Authors further respond by stating that neither this manuscript and nor Lin et al. 2022 observed any gene expression changes that are specific to SSCs. However, this statement only partly answers my question. Since this manuscript is coming after Lin et al. 2022 and is specifically dealing with DOT1L mediated changes in other chromatin features, it is therefore expected to provide some level of advancement over the existing literature in this context. If no significant change is observed in relation to defects displayed by SSCs, it might be more appropriate to state that rather than just making a correlation with the previous data. Other than this, all other comments have been properly addressed.

Point-by-point response to the reviewers' comments.

We want to thank the Reviewers for their constructive evaluation of our manuscript which helped us improve the presentation and interpretation of our data. We have now thoroughly revised the manuscript based on their suggestions and requests. Point-by-point responses to Reviewer comments can be found below (in blue).

Reviewers' comments:

Reviewer #1 (Remarks to the Author):

Coulee et al investigated the distribution of H3K79me2 across different steps of male germ cell development and the effect of DOT1L knockout on gene expression at these different developmental steps. They also integrate their data with publicly available data of other histone modifications and chromosome conformation. The authors found changes in H3K79me2, both on protein-coding genes and enhancers, during the male germ cell development. DOT1L knockout results in a limited but increasing number of genes being deregulated over the developmental steps, with a trend towards upregulation. Finally, the authors found a difference between how autosomal and sex chromosomes respond to DOT1L loss, with the latter associated with downregulation of genes. The study is of interest but remains generally descriptive and several of the conclusions made by the authors are, in my opinion, not supported by what they are currently showing in the manuscript.

Comments:

1. I think the statements on the role of DOT1L in inhibiting gene expression should be tone down. Knockout models are notoriously known to be associated with compensatory effects that make it difficult to conclude on the direct functions of a protein. To prove an inhibitory function of DOT1L would require at least experiments to show that DOT1L is binding to these repressed genes and that a rapid degradation/inhibition results in the direct transcriptional activation of these genes.

DOT1L ChIP-seq experiments would indeed be informative, and we tried for several months to perform anti-DOT1L ChIP-qPCR/Seq experiments. We tested different antibodies without success. We think that the available anti-DOT1L antibodies do not work for ChIP in mouse samples. This observation was also made by colleagues in other labs, and, to the best of our knowledge, there is no DOT1L ChIP-seq data from mouse cells in the literature (only from human cells, using an antibody that does not recognize the mouse version).

Regarding experiments of rapid degradation/inhibition of DOT1L: Since it is impossible to recapitulate a functional spermatogenesis *in vitro*, addressing this point would require a technically challenging *in vivo* approach.

It is worth noting that our results are consistent with previous publications using other knockout models in which there were more upregulated than downregulated genes, concluding on a repressive effect of DOT1L (reviewed in (Wille and Sridharan, 2022)).

We agree that a general limitation of RNAseq analyses is the inability to distinguish between direct and indirect effects. However, whether direct or indirect, the effect is indeed due to *Dot1l* loss. To address the reviewer's remark and clarify our intent, we have slightly modified the results section title to “Spike-in RNA-seq analyses reveal that DOT1L has a repressive effect on gene expression” instead of “Spike-in RNA-seq analyses reveal that DOT1L has a broad repressive rather than activating effect on gene expression”.

2. From the authors RNA-seq data, are the upregulated genes found in DOT1L-KO condition activated (no expression in control and expression in DOT1L-KO), upregulated (expressed to some extent in control but more expressed in DOT1L-KO), or a mix of the two? It would have been interesting to know whether these upregulated were also associated with paused RNA polymerase (RNAP) II or are totally devoid of RNAPII but it does not seem that data are available to conclude about this.

The analysis of the number of deregulated genes according to their level of expression in control condition reveals that upregulated genes are mostly low expressed ones but not exclusively (see Figure 1 below). Conversely, the distribution of downregulated genes is more of a Gaussian shape in the control condition.

Response to reviewer Figure 1. Distribution of deregulated genes according to their level of expression in the control condition. $X = \text{percentiles of level of expression}$, $Y = \text{number of genes}$.

We analyzed RNAPII binding at the promoters of deregulated genes based on available datasets (see Figure 2 below). This shows that the level of RNAPII binding in promoters of upregulated genes is very low compared to downregulated genes, which is consistent with the difference in levels of expression shown in the figure above. However, we cannot distinguish between paused or elongating RNAPII since the antibody targets the total RNAPII.

Analyzing the status of RNAPII at those genes would still require further experiments which are out of scope of this study.

Response to reviewer Figure 2. RNAPII binding profiles at genes deregulated following *Dot1l*-knockout in primary spermatocytes (left panel) and in round spermatids (right panel) DR: downregulated, UR: upregulated. (References of published datasets of RNA pol II ChIP-seq used in the figure: <https://www.ncbi.nlm.nih.gov/geo/query/acc.cgi?acc=GSE81470> and <https://www.ncbi.nlm.nih.gov/geo/query/acc.cgi?acc=GSE45441>).

3. Could the increase in gene deregulation over the male germ cell development be explained by the deregulation of a few key genes at the start of the male germ cell development? Any defect from the beginning could continue to expand independently of any effects associated with the loss of DOT1L.

In theory, this would be possible. However, few genes are deregulated at premeiotic stages and none of them is a known regulator of gene expression. Besides, the genes found deregulated in premeiotic male germ cells are not found deregulated afterwards, in meiotic and postmeiotic cells. Hence, we do not have evidence for this.

4. From Blanco et al, 2023 (previous paper from the research group), H3K79me2 has disappeared at the RS developmental step but there is some low level left at the previous SC step. Do the authors know how much H3K79me2 is left in GSG? More than in SC? What would be their expectation for the levels of H3K79me1 and H3K79me3?

Based on our experience and on data from previous studies (for review see (Vlaming and van Leeuwen, 2016)), we expect H3K79me1 and H3K79me3 to be reduced similarly to H3K79me2. For instance, we found a similar reduction in the level of H3K79me1 and H3K79me2 by LC-MS/MS in *Dot1l*-KO whole testes, round spermatids, elongating spermatids and spermatozoa (Blanco et al., 2023). H3K79me3 was too weak to be detected by this approach but was similarly decreased by immunofluorescence. Regarding the level of expression of H3K79me2, we provide new western blot images of H3K79me2 in *Dot1l*-KO and control (CTL) germinal stem cells (GSC) in the revised version of this manuscript (revised Figure S4).

5. As the DOT1L KO condition should result in the loss of most, if not all, H3K79 methylation, the fact that only up to ~1,000 genes are affected at most in one condition more than 1.5-fold seems

rather to indicate that H3K79 methylation is generally not really required for controlling gene expression.

We agree with the reviewer that although our results show a global upregulation of autosomal genes in KO compared to CTL meiotic and postmeiotic cells following DOT1L loss, the magnitude of this increase is relatively modest (see Figure 3 below). This is in line with previous results in somatic cells (reviewed in (Wille and Sridharan, 2022)). To clarify, we have added the following figure (revised Fig 3d) and modified the text in the Results section and in the Discussion.

Lines 209-211 in the clean version of the revised MS: “This observation was confirmed by studying the distribution of gene expression level in meiotic and postmeiotic cells: the average expression of autosomal genes was significantly higher in KO vs. CTL in SCI, SCII and RS (Fig 3d).”

Lines 474-475 in the clean version of the revised MS: “While this upregulation appears widespread, it is of small intensity. This indicates that DOT1L is unlikely involved in the primary control of gene expression but rather in its modulation.”

Response to reviewer Figure 3. Log₁₀(cpm) expression of all genes in CTL or Dot1l-KO (KO) sample expression in SCI, SCII and RS. Adjusted p-values using Wilcoxon test adjusted with Benjamini-Hochberg correction are indicated above each KO vs CTL comparison.

6. It is the same issue with the X chromosome, yes most of the downregulated are on the X chromosome but compared to all the genes expressed on the X chromosome, what is the proportion downregulated in the DOT1L KO condition? It looks like at most a few percent of the genes so I am not really convinced DOT1L plays as written in the Discussion a pivotal role in XY gene activation (with point 6 below, it looks like more X chromosome only rather than XY).

We agree with the reviewer that the majority of expressed X-linked genes are not significantly deregulated. This is also observed in the above figure (Response to reviewer Figure 3 and revised Figure 3d). We have modified the Result and Discussion sections of our manuscript accordingly.

Lines 329-332 in the clean version of the revised MS: “No significant difference in gene expression level was observed between KO and CTL samples for X-linked genes, indicating that the over-representation of X-linked genes among downregulated genes in SCII and RS applies only to a subset of X-linked genes (Fig 3d).”

Yet, the difference in the direction of deregulation between autosomal genes and X-linked genes, is striking and significant. If we look at all deregulated genes with a non-adjusted p-value, we observe a significant decrease in the fold change expression between KO and control of X-linked genes compared to autosomal genes in SCI and RS (see figure below and revised Figure 5 in the manuscript). Moreover, when we analyze the level of expression of all X-encoded genes and autosomal genes, the fold change between KO and control is still significantly lower for X-linked genes than autosomal genes in all meiotic and postmeiotic stages (see figure below and revised Figure S5 in the manuscript). Importantly, genes on the Y chromosome behave differently and their fold change of expression is significantly higher compared to autosomal genes (and to X-linked genes).

Response to reviewer Figure 4. Gene expression difference between *Dot1l*-KO and control, at different spermatogenesis cell stages according to the type of chromosomes. Boxplots show Log₂-fold changes of *Dot1l*-KO relative to CTL gene expression according to the chromosome location, in each cell type. The left panel shows log₂FC values for differentially expressed genes with a p-value < 0.05. The right panel shows Log₂FC values for all genes detected in the RNA-seq analyses (including not significantly deregulated). Box: 25th/75th percentiles. Bar in the box: median. Whiskers: 1.5 times the interquartile range from the 25th/75th percentiles. Stars indicate a significant FDR calculated using Wilcoxon test adjusted with Benjamini-Hochberg correction (*, p < 0.05; **, p < 0.005; ***, p < 0.0005; ****, p < 0.00005; ns, not significant).

(NB. These figures are also shown in Figures 5 R1 and S5 R1 of the revised version of the manuscript).

7. Rather than sex chromosomes, Figure 5 seems to show that it is the X chromosome only that is associated with downregulated genes in the DOT1L-KO condition.

We thank the reviewer for this observation and agree on it. As mentioned in our reply to question 6, the downregulation indeed only concerns X-linked genes. We have changed the wording in the manuscript and propose a new title. Figure 5b has also been changed to show separate analyses for X and Y chromosomes as shown in the figure above (which corresponds to Figure 5 R1 and S5 R1). We have added a paragraph addressing Y-linked gene deregulation (see reply to point 14 from reviewer 2 below).

8. Why did the authors choose H3K79me2 rather than H3K79me1 or H3K79me3 that are also dependent on DOT1L activity? This should be explained in the manuscript.

As mentioned in response to point 4, during normal spermatogenesis, H3K79me1 shows a low and unchanged level of enrichment during meiosis while H3K79me2/3 increase after meiosis and before histone hyperacetylation (preceding the histone to protamine transition), suggesting that they may play a role in this process. We decided to focus on H3K79me2 because most publications in cell types other than germ cells have studied this mark, which facilitated the comparison of results. These explanations have been added at the beginning of the Results section of the revised version of our manuscript:

Lines 85-88 in the clean version of the revised MS: "...we focused on H3K79me2 (instead of H3K79me1 or H3K79me3) because it is both dynamic and abundant in male germ cells, as well as the most studied mark in other cell types, facilitating the comparison of results. "

9. Several histone marks (H3K4me3, H3ac/H4ac, H3K36me3, H3K79me, and H3K9me3/H3K27me3 for low/no expression) can be generally seen as a proxy for RNAPII transcriptional activity. I am wondering if it would not have been more informative to have performed a RNAPII ChIP-seq in addition/rather than H3K79me2. This would have provided a lot more information on gene expression and on RNAPII initiation/pausing and elongation on protein-coding genes and eRNA activity +/- DOT1L knockout.

Our results and several studies (see for instance (Aslam et al., 2021; Godfrey et al., 2019; Kwesi-Maliepaard et al., 2020)) suggest an interplay between H3K79me2 and other histone modifications. Hence, we decided to explore the chromatin environment rather than RNAPII positioning. In the revised version of our manuscript, we confirm the model derived from our *in silico* analyses by performing quantitative CUT&Tag on *Dot11*-KO and CTL round spermatids (see revised Figure 4 and associated paragraph below).

Lines 286-299 in the clean version of the revised MS: "To test the accuracy of this model, we next performed quantitative (spike-in) CUT&Tag experiments for H3K27ac and H3K27me3 in *Dot11*-KO and control RS, in triplicates (Fig S1c, Fig 4f and 4g). Normalized enrichment profiles showed that, in control RS, genes upregulated in the KO have low levels of H3K27ac and very high levels of H3K27me3 at their promoter/TSS. Upon *Dot11* loss, their enrichment in H3K27me3 decreased while H3K27ac slightly increases, consistent with their upregulation. Conversely, in control RS, downregulated genes have higher levels of H3K27ac and are almost devoid of H3K27me3. Upon *Dot11* loss, their H3K27ac signal was visibly reduced while H3K27me3 remains unchanged, consistent with their downregulation. Finally, genes unaffected by *Dot11*-KO have intermediate H3K27me3 levels (lower than upregulated genes) and low H3K27ac enrichment at their promoter/TSS (Fig 4f and g). Altogether, these findings confirm the model derived from the ChromHMM analyses and show that the impact of DOT1L on gene expression depends on the local chromatin environment of these genes, particularly on H3K27me3, H3K27ac, and/or H3K79me2 enrichment (see Fig 6)."

We think that our work provides novel and interesting information on H3K79me2 dynamics during spermatogenesis, on its relation to the transcriptional remodeling during this process, and on the chromatin environment of genes deregulated in *Dot1l*-KO male germ cells.

Reviewer #2 (Remarks to the Author):

Reviewer comments for Coulee et al. 2024:

Manuscript by Coulee et al., entitled “DOT1L has opposite effects on sex chromosome and autosomal gene expression in male germ cells” describes the molecular role of DOT1L in transcriptional regulation of male germ cell development. The manuscript builds on the previous publication (Blanco et al. 2023) and demonstrates in more detail the transcriptional changes associated with DOT1L loss in mal germ cells. The major finding of the manuscript is that DOT1L influences the transcriptional outcome by different mechanisms depending on the chromatin environment and chromosomal location. Authors conclude that DOT1L specifically activates genes that are located on sex (XY) chromosomes but represses genes that are present on autosomes. Authors further argue that the regulation on sex chromosomes is directly mediated via H3K79me2, whereas on autosomes DOT1L acts indirectly by influencing the balance between H327me3 and H3K27ac chromatin on gene promoters and enhancers. While the gene expression changes and genomic features of H3K79me2 are properly established, some of the results are over concluded and no clear mechanistic insight is provided that can clearly explain differential role of DOT1L on autosomal genes and sex-linked genes. Also, the finding that DOT1L represses some genes and activates others is not novel and has been reported before. Nevertheless, the current manuscript presents a big advancement by including the H3K79me2 ChIP-seq data characterization from different types of germ cells and therefore can be of interest to other researchers in the field. Therefore, for the manuscript to be considered for publication, the following points should be properly addressed.

Major comments:

1) Previous literature on role of DOT1L in male germ cell development indicates that DOT1L acts at two different stages in spermatogenesis: 1) at mitotic proliferation of spermatogonial stem cells, and 2) postmeiotic spermatid differentiation (Lin et al., *Genes Dev.*, 2022; Lin et al., *Dev* 2023; Blanco et al., *EMBO Rep*, 2023; and Malla et al., *Dev* 2023). However, the data presented in the current study (Fig 1 and Fig 2c) suggests that majority of H3K79me2 enriched peaks (91%) in GSC are associated with transcriptionally weak or quiescent chromatin (lines 154-157), implying that DOT1L deletion should have a minimal impact on gene expression in (Kit-) undifferentiating spermatogonia (Fig 3b). This does not support the already established role for DOT1L in spermatogonial stem cell (SSC) proliferation and maintenance. Authors should explain why they do not observe gene expression changes specific to SSCs.

The number of DEGs in Kit- (and Kit+) is indeed small but this is in line with previously published studies. Indeed, (Lin et al., 2022) also found a low number of deregulated genes in premeiotic stages (38 DEG). Their study also reveals that the small number of deregulated genes when DOT1L is inhibited does not preclude a significant effect on these cell's phenotype. The fact that

H3K79me2 is mostly associated with weak/quiescent chromatin at this stage is also consistent with the small number of genes that are deregulated in the knockout.

In addition, the GO analysis of DEGs in Lin et al. showed enrichment in processes such as “anterior/posterior pattern specification”, “proximal/distal pattern formation” and “sequence-specific DNA binding”, rather than SSC-specific processes.

2) Fig 2d shows that enhancers that are enriched in H3K79me2 promote nearby gene expression compared to H3K79me2 depleted enhancers (lines 177-180). But to conclude that H3K79me2 imparts stronger activity to H3K79me2 enriched enhancers, authors need to show that the enhancer activity is altered in the absence of DOT1L by comparing the enrichment of H3K4me1 and/or H3K27ac on some of these enhancer regions (by ChIP-qPCR).

The aim of these analyses was to check if we observe a similar association between H3K79me2 at enhancers and higher level of expression in male germ cells as observed in somatic cells by other groups (Cattaneo et al., 2022; Godfrey et al., 2019), and indeed we show that H3K79me2 at gene bodies and distal enhancers is a marker of active transcription.

To address the specific question of H3K27ac level at enhancers, we plotted the enrichment profiles of spike in H3K27ac CUT&Tag that we performed in triplicates KO and CTL RS. The results shown in the figure below indicate that, in CTL RS, the level of H3K27ac at enhancers enriched in H3K79me2 (+H3K79me2; right plot, blue lines) is higher than for enhancers devoid of H3K79me2 (- H3K79me2; left plot, blue lines). In *Dot1l*-KO RS, H3K27ac levels decrease at Enh+H3K79me2.

Response to reviewer Figure 5. Level of H3K27ac enrichment at genic enhancers devoid of H3K79me2 (left plot) and enriched in H3K79me2 (right plot) in *Dot1l*-CTL and *Dot1l*-KO round spermatids.

3) In Fig 4b, authors demonstrate that the enhancers of upregulated genes are enriched in H3K79me2 and conclude that the repressive effect of DOT1L on gene expression is independent of H3K79me2 (lines 259 - 262). But this conclusion could be drawn only for genes that are completely devoid of H3K79me2 and not for the genes whose expression is driven by H3K79me2 enriched enhancers. Furthermore, if authors suggest that the gene upregulation effect observed in *Dot1l*KO germ cells is guided by H3K79me2 enriched enhancers, this could mean that H3K79me2

in the wild type cells is repressing these enhancers. Authors should define role of H3K79me2 at these enhancers and explain how they contribute to gene regulation in Dot11KO cells.

We thank the reviewer for this remark. We have reworded our sentence as follows:

“Upregulated genes are, therefore, mostly either devoid of H3K79me2 or enriched in H3K79me2 strictly at nearby enhancers and rarely at their gene body.” (lines 256-257 in the clean version of the revised MS).

We have now clarified the model we believe applies to DOT1L in male germ cells, in an additional figure (new Figure 6). This model is derived from our analyses of the local chromatin environment of gene (promoter/TSS) (see reply to point 4 below and Response to reviewer Figure 5).

4) Based on Fig 4a, Authors conclude that H3K79me2 is more enriched in genes that are not deregulated than the differentially regulated genes (lines 243-246). This is problematic because it contradicts the existing notion in the field that H3K79me2 is positively associated with transcriptional regulation.

Our data (and others) show that H3K79me2 labels actively transcribed genes, yet the majority of H3K79me2+ genes in WT are not deregulated when DOT1L (and H3K79me2) is lost. This agrees with the recent literature on the topic (reviewed in (Wille and Sridharan, 2022)).

Also, the conclusion that most of H3K79me2-enriched genes do not exhibit any change in expression in Dot11 deleted cells could be interpreted to mean that the downregulation following DOT1L loss itself is not specific. Authors should explain how H3K79me2 loss can be selectively impairing gene expression at some genes (downregulated genes) and not others (unaltered genes).

We performed quantitative (spike-in) CUT&Tag experiments using H3K27ac and H3K27me3 antibodies in round spermatids. These analyses show that, in wild-type (control) round spermatids, downregulated genes (and X-linked genes in general) are in an H3K27me3-poor environment, upregulated genes display particularly high levels of H3K27me3 at their promoters, while autosomal (not deregulated genes) show intermediate levels of H3K27me3. In *Dot11*-KO round spermatids, H3K27me3 is decreased at the promoter of upregulated genes and unchanged for other genes.

H3K27ac enrichment profiles are also informative: in wild type round spermatids, H3K27ac level is high at the promoter of down regulated genes and autosomal (not deregulated genes) and very low at upregulated genes. In *Dot11*-KO, H3K27ac is slightly increased at the promoter of upregulated genes, and decreased at downregulated and X-linked genes.

In sum, downregulated genes and unaltered genes both carry H3K79me2 and what distinguishes them is the lack of H3K27me3. In contrast, the chromatin signature of upregulated genes is a high H3K27me3 environment and a low H3K79me2 signal.

These observations confirm that the impact of DOT1L on gene expression depends on their local chromatin environment, and more specifically on their enrichment in H3K79me2 and H3K27me3 (and in H3K27ac).

We added a model figure to illustrate our findings (Figure 6 and graphical abstract), a new paragraph in the results section (lines 286-299 in the clean version of the revised MS) and updated Figure 4 to include these new analyses (revised Fig 4f and 4g):

“To test the accuracy of this model, we next performed quantitative (spike-in) CUT&Tag experiments for H3K27ac and H3K27me3 in Dot11-KO and control RS, in triplicates (Fig S1c, Fig 4f and 4g). Normalized enrichment profiles showed that, in control RS, genes upregulated in the KO have low levels of H3K27ac and very high levels of H3K27me3 at their promoter/TSS. Upon Dot11 loss, their enrichment in H3K27me3 decreased while H3K27ac slightly increases, consistent with their upregulation. Conversely, in control RS, downregulated genes have higher levels of H3K27ac and are almost devoid of H3K27me3. Upon Dot11 loss, their H3K27ac signal was visibly reduced while H3K27me3 remains unchanged, consistent with their downregulation. Finally, genes unaffected by Dot11-KO have intermediate H3K27me3 levels (lower than upregulated genes) and low H3K27ac enrichment at their promoter/TSS (Fig 4f and g). Altogether, these findings confirm the model derived from the ChromHMM analyses and show that the impact of DOT1L on gene expression depends on the local chromatin environment of these genes, particularly on H3K27me3, H3K27ac, and/or H3K79me2 enrichment (see Fig 6).”

5) Fig 4c and 4e. Authors conclude that upregulated genes are enriched in H3K27me3, and downregulated genes are enriched in H3K4me3 states in the wildtype cells (lines 280 – 283 and 443 - 449). This indicates that DOT1L somehow impacts deposition of H3K27me3 on genes that it represses, and that the gene upregulation observed in Dot11KO cells is due to a shift from H3K27me3 chromatin to H3K27ac chromatin at the deregulated genes (lines 451-452). But no H3K27me3 or H3K27ac enrichment in the Dot11KO cells is shown. Authors should therefore provide H3K27me3 and H3K27ac enrichment (by ChIP-qPCR) at some of the DEGs in Dot11KO cells to test if these modifications are differentially regulated in Dot11KO germ cells. This data is needed to establish the H3K79me independent role for DOT1L at genes that are upregulated upon its loss.

We thank the Reviewer for this comment. As described above (reply to point 4 and response to reviewer Figure 6), we performed quantitative CUT&Tag experiments for both H3K27me3 and H3K27ac marks on KO and CTL round spermatids. These analyses confirm our model that upregulated and downregulated genes are in distinct local chromatin environment which are modified upon Dot11 loss. See also result section, lines 286-299 of the clean version of the revised MS and revised Figure 4.

Minor comments:

6) There is a typo correction in line 96. it should be as..."iii) round spermatids (RS), and iv) elongating spermatids/condensed spermatids (ES) as postmeiotic..."

This has been corrected.

7) Figures are presented in a nonuniform alphabet case. Some figure panels are presented with

upper case letters such as in Fig. 3 and Fig. 5, and others with lower case such as Fig.1, Fig.2 and Fig. 4. This should be corrected.

Thank you for your remark. Figures and captions have been harmonized.

8) According to Fig 2B and 2C, Quiescent chromatin states are represented by both H3K79me2 enriched and H3K79me2 devoid regions. Can authors comment on what it means in terms of their gene expression regulation by H3K79me2 or upon *Dot1l* deletion.

Quiescent chromatin states have low H3K27ac and H3K27me3 levels. The genes located in these regions (whether H3K79me2+ or -) are not deregulated upon *Dot1l* loss, and this is in agreement with the model that we present in the revised version of our manuscript (see also our replies above and revised figure 6).

9) In lines 161-166 (page 6), authors mention that 70% of H3K79me2 enriched regions that are bivalent were specifically gained in round spermatids compared to 34% in SCI. What could be the impact of DOT1L loss (or loss of H3K79me2) on these bivalent regions with respect to their gene expression. Based on their RNA Seq data, authors should demonstrate whether there is any change in the expression of poised chromatin in SCI and RS when DOT1L depleted.

To address this specific question, we show in the figure below the mean expression of genes located in bivalent regions (with or without H3K79me2). The statistical analysis shows a significant upregulation of those genes upon *Dot1l* loss, consistent with our model. The upregulation is particularly striking for genes which are in a H3K79me2 poor environment, which is consistent with our model (see revised figure 6).

Response to reviewer Figure 5. Mean expression of genes located in bivalent chromatin environment devoid (-H3K79me2) or enriched (+H3K79me2) in H3K79me2. Data are shown for *Dot1l*-KO and control RS. Box: 25th/75th percentiles. Bar in the box: median. Whiskers: 1.5 times the interquartile range from the 25th/75th percentiles. Dashed lines: log₂(1.5) fold change. The p-value calculated using Wilcoxon test adjusted with Benjamini-Hochberg correction are shown above each comparison.

10) Authors should provide a detailed protocol for the isolation of different testicular cell types by flow cytometry. Authors should clearly indicate how were secondary spermatocytes (SCII) distinguished from other diploid cells like Sertoli cells and other somatic cells. Also, authors should provide a western blot image for the specific markers of each cell type to confirm the percent purity of separation for each cell type that was isolated by either cell sorting or elutriation.

Regarding FACS, the two-steps method to prepare cell suspension, together with the use of the somatic β 2m marker, allows the removal of testicular somatic cells. To identify and confirm the purity of the meiotic populations in the Hoechst profile of testicular cells, we have previously used different well-defined models of early spermatogenesis block (*W/W^o* mutant and cryptorchid males for example), in which meiotic cells are completely absent. These data and microscopic analysis of sorted cells show that the Hoechst profile of testicular cells defines a population of spermatocytes II that is distinct from other diploid cells (Bastos et al., 2005; Lassalle et al., 2004). It has been previously shown that this diploid population presents the transcriptomic signature of meiotic spermatocytes II (Bastos et al., 2005), and for present data, see below).

In the revised version of our MS, we provide more details in the M&M section to clarify this point (see lines 596-606 in the clean version of the revised MS). We also add a heatmap obtained from the present RNA-seq dataset, which shows the expression level of markers of Sertoli cells (the predominant somatic cells in the testis) along with that of male germ cell markers (see Figure S4c R1). It confirms the specificity of our data and the absence/very low level of somatic cell contamination.

11) According to Fig 3B, the effect of DOT1L loss upon gene expression is greater in secondary spermatocytes (SCII) (more DEGs) than primary spermatocytes (SCI) or round spermatids (RS). Therefore, relative enrichment of H3K79me2 in SCII should be shown (by immunohistochemistry).

To address this question, we add, in the revised version of the manuscript, Western blot images from *Dot1l*-KO and control SCI, SCII and RS, detected with anti-H3K79me2 antibodies (Figure S4 R1). This panel shows that H3K79me2 and DOT1L decrease in KO SCII is higher than in SCI and similar to what is observed in RS.

Immunofluorescence/IHC images of H3K79me2 staining in *Dot1l*-KO vs. control testes can be found in our previous article (Blanco et al., 2023).

12) Authors mention that most of the genes that are deregulated in Kit⁺ spermatogonia are not differentially regulated in meiotic or postmeiotic germ cells (lines 222-224). Can authors comment on Why that is the case. Is it because these genes are not expressed in meiotic and postmeiotic stages.

Commented [J1]: Vérifier que tu as la bonne version de S4

The chromatin and transcriptional landscape of male germ cells is extremely dynamic between spermatogonia and meiotic/postmeiotic cells, as shown by other studies (see for instance (Soumillon et al., 2013)) and single cell analyses (Chen et al., 2018; Ernst et al., 2019; Green et al., 2018)) and illustrated in Figure S2 of our manuscript. This, indeed, may explain the fact that deregulated genes in spermatogonia are not deregulated at later stages.

13) In Figs 5a and 5b, authors report that sex-linked genes are over-represented in downregulated gene list in Dot11KO cells (lines 306-315). However, these sex-linked genes are also significantly down regulated in primary spermatocytes when MSCI is inactive and sex chromosome transcription is inhibited (Fig. 5b). Ideally there should be no change in sex linked genes in primary spermatocytes (SCI) since these are naturally turned off due to MSCI. Authors should comment on why sex-linked genes are still down in SCI.

We thank the reviewer for this comment. The original figure 5b was comparing Log2FC levels from autosomal vs XY linked genes. We have performed additional analyses to specifically address the question of (absolute not relative) global level of X gene expression in KO and CTL cells. The results are presented in Figure 3d R1 (see also above **response to reviewer Figure 3**) and show that the mean level of expression of X genes is not significantly different in KO vs CTL, in SCI as well as in all other cell types. However the mean level of expression of autosomal genes is significantly higher in KO vs CTL, in SCI, SCII and RS. In Fig 5b, we compared the Log2FC of KO vs CTL expression in autosomes vs. sex chromosomes and found it is significantly different in almost all cell stage. Because of Figure 3d, we now think it is both because of the upregulation of autosomal gene expression and the downregulation of X expression in the KO. We have added the following sentence to explain it:

Lines 328-331 in the clean version of the revised MS: “No significant difference in gene expression level was observed between KO and CTL samples for X-linked genes, indicating that the over-representation of X-linked genes among downregulated genes in SCII and RS applies to a subset of X-linked genes (Fig 3d).”

Figure3d_R1 also shows that MSCI is visible in KO and CTL SCI, as indicated by the very low level of X and Y gene expression in SCI compared to autosomes.

14) Highest enrichment of H3K79me2 is detected at Y chromosome yet highest gene down regulation is observed on X chromosome (Fig 5d). According to this data, if sex chromosome gene activity is directly dependent on de novo H3K79me2, as is suggested, then the Y linked genes should exhibit highest down regulation.

We have now performed new separate analyses for X and Y genes, which revealed a different pattern of expression and deregulation between these two chromosomes. See revised figures 3d, 5b and S6 as well as a modified paragraph lines 337-348 in the clean version of the revised MS:

“Surprisingly, chromosome Y-linked gene expression in Dot11-KO displayed a different profile with a slight but significant upregulation in SCII (Fig 3d). By RNA-seq, however, few to no Y-linked genes were found significantly deregulated, except Zfy1, found downregulated and pseudogenes corresponding to the multicopy genes Ssty and Sly, which were upregulated in SCII

(Table S2). By qRT-PCR performed in RS, we confirmed the down regulation of *Zfy1* and the upregulation of *Sly1* but not of *Ssty1* or *Ssty2* (Fig S6c). *Slx* and *Slx11*, the X-linked homologs of *Sly*, were not deregulated while, the downregulation of the X-encoded single copy genes *Ube2a* and *Hdac8* was confirmed. It is worth noting that *Sly* and *Ssty* are present in >100 copies and represent the vast majority of Y-linked genes. They are only expressed in SCII and RS; their upregulation could therefore explain the global upregulation of Y gene expression observed in SCII and in RS (Fig 3d, 5b, and S6b). These observations point to a complex role of DOT1L in the regulation of Y chromosome-encoded genes.”

15) Fig S8 is provided but not mentioned anywhere in the text. It should be removed or discussed in the text.

Fig S8 refers to the method to characterize chromatin states and is now discussed in the M&M section.

16) Authors propose that DOT1L mediates gene repression by limiting H3K27 acetylation (and hence promoting H3K27me3 deposition) via expression of *Hdac8* (lines 455-460). Authors should explain how downregulated *Hdac8* in *Dot11*KO cells is specifically acting on autosomal genes only and not acting on sex linked genes that are downregulated.

We hope that the additional experiments and figures we present in the revised version of our manuscript clarify our model. We propose that the direction of deregulation observed in *Dot11*-KO depends on the chromatin environment. In male germ cells, X chromosome and autosomal genes have a different chromatin environment, in particular, sex chromosomes are notably devoid of H3K27me3 (Moretti et al., 2016; Sin et al., 2015). We mentioned HDAC8 along with other repressors to illustrate that gene expression regulation in the present model is certainly more complex than strictly defined by the chromatin marks we studied. We do not know the underlying mechanism by which HDAC8 could impact only autosomal genes. Could it be influenced by the chromatin environment and/or by other repressors? In the revised version of our manuscript we also discuss the observation that *Sly*, which encodes a known repressor of XY gene expression is upregulated in *Dot11*-KO SCI and RS, further illustrating the complexity of the regulation of gene expression in the present system.

17) Authors discuss that *Kdm6a* is a H3K27 methylase; and that it is downregulated in *Dot11*KO Secondary spermatocytes and round spermatids (line 457-460). Firstly, *KDM6A* is not a H3K27 methylase but a H3K27me3 demethylase. Authors should correct that. Secondly, down regulation of *KDM6A* should lead to enrichment of H3K27me3 in *DOT1L*KO cells since it is a H3K27 demethylase. The conclusion that there is a shift from H3K27me3 based chromatin to H3K27ac chromatin in *Dot11*KO cells does not make sense in the context of the proposed model.

We apologize for this oversight, we have corrected methylase for demethylase. We agree with the reviewer: Indeed, what we meant in this sentence was that the model is not straightforward, since we observe the downregulation of repressors (such as HDAC8) in parallel with the downregulation of an activator (*KDM6A*). We have modified the sentence to clarify its meaning.

References

- Aslam, M.A., Alemdehy, M.F., Kwesi-Maliepaard, E.M., Muhaimin, F.I., Caganova, M., Pardieck, I.N., van den Brand, T., van Welsem, T., de Rink, I., Song, J.-Y., de Wit, E., Arens, R., Jacobs, H., van Leeuwen, F., 2021. Histone methyltransferase DOT1L controls state-specific identity during B cell differentiation. *EMBO Rep* 22, e51184. <https://doi.org/10.15252/embr.202051184>
- Bastos, H., Lassalle, B., Chicheportiche, A., Riou, L., Testart, J., Allemand, I., Fouchet, P., 2005. Flow cytometric characterization of viable meiotic and postmeiotic cells by Hoechst 33342 in mouse spermatogenesis. *Cytometry A* 65, 40–49. <https://doi.org/10.1002/cyto.a.20129>
- Blanco, M., El Khattabi, L., Gobé, C., Crespo, M., Coulée, M., de la Iglesia, A., Ialy-Radio, C., Lapoujade, C., Givélet, M., Delessard, M., Seller-Corona, I., Yamaguchi, K., Vernet, N., Van Leeuwen, F., Lermine, A., Okada, Y., Daveau, R., Oliva, R., Fouchet, P., Ziyat, A., Pflieger, D., Cocquet, J., 2023. DOT1L regulates chromatin reorganization and gene expression during sperm differentiation. *EMBO Rep* 24, e56316. <https://doi.org/10.15252/embr.202256316>
- Cattaneo, P., Hayes, M.G.B., Baumgarten, N., Hecker, D., Peruzzo, S., Aslan, G.S., Kunderfranco, P., Larcher, V., Zhang, L., Contu, R., Fonseca, G., Spinozzi, S., Chen, J., Condorelli, G., Dimmeler, S., Schulz, M.H., Heinz, S., Guimarães-Camboa, N., Evans, S.M., 2022. DOT1L regulates chamber-specific transcriptional networks during cardiogenesis and mediates postnatal cell cycle withdrawal. *Nat Commun* 13, 7444. <https://doi.org/10.1038/s41467-022-35070-2>
- Chen, Y., Zheng, Y., Gao, Y., Lin, Z., Yang, S., Wang, T., Wang, Q., Xie, N., Hua, R., Liu, M., Sha, J., Griswold, M.D., Li, J., Tang, F., Tong, M.-H., 2018. Single-cell RNA-seq uncovers dynamic processes and critical regulators in mouse spermatogenesis. *Cell Res* 28, 879–896. <https://doi.org/10.1038/s41422-018-0074-y>
- Ernst, C., Eling, N., Martinez-Jimenez, C.P., Marioni, J.C., Odom, D.T., 2019. Staged developmental mapping and X chromosome transcriptional dynamics during mouse spermatogenesis. *Nat Commun* 10, 1251. <https://doi.org/10.1038/s41467-019-09182-1>
- Godfrey, L., Crump, N.T., Thorne, R., Lau, I.-J., Repapi, E., Dimou, D., Smith, A.L., Harman, J.R., Telenius, J.M., Oudelaar, A.M., Downes, D.J., Vyas, P., Hughes, J.R., Milne, T.A., 2019. DOT1L inhibition reveals a distinct subset of enhancers dependent on H3K79 methylation. *Nat Commun* 10, 2803. <https://doi.org/10.1038/s41467-019-10844-3>
- Green, C.D., Ma, Q., Manske, G.L., Shami, A.N., Zheng, X., Marini, S., Moritz, L., Sultan, C., Gurczynski, S.J., Moore, B.B., Tallquist, M.D., Li, J.Z., Hammoud, S.S., 2018. A Comprehensive Roadmap of Murine Spermatogenesis Defined by Single-Cell RNA-Seq. *Dev Cell* 46, 651–667. <https://doi.org/10.1016/j.devcel.2018.07.025>
- Kwesi-Maliepaard, E.M., Aslam, M.A., Alemdehy, M.F., van den Brand, T., McLean, C., Vlaming, H., van Welsem, T., Korthout, T., Lancini, C., Hendriks, S., Ahrends, T., van Dinther, D., den Haan, J.M.M., Borst, J., de Wit, E., van Leeuwen, F., Jacobs, H., 2020. The histone methyltransferase DOT1L prevents antigen-independent differentiation and safeguards epigenetic identity of CD8+ T cells. *Proceedings of the National Academy of Sciences* 117, 20706–20716. <https://doi.org/10.1073/pnas.1920372117>

- Lassalle, B., Bastos, H., Louis, J.P., Riou, L., Testart, J., Dutrillaux, B., Fouchet, P., Allemand, I., 2004. "Side Population" cells in adult mouse testis express *Bcrp1* gene and are enriched in spermatogonia and germinal stem cells. *Development* 131, 479–487. <https://doi.org/10.1242/dev.00918>
- Lin, H., Cheng, K., Kubota, H., Lan, Y., Riedel, S.S., Kakiuchi, K., Sasaki, K., Bernt, K.M., Bartolomei, M.S., Luo, M., Wang, P.J., 2022. Histone methyltransferase DOT1L is essential for self-renewal of germline stem cells. *Genes Dev* 36, 752–763. <https://doi.org/10.1101/gad.349550.122>
- Moretti, C., Vaiman, D., Tores, F., Cocquet, J., 2016. Expression and epigenomic landscape of the sex chromosomes in mouse post-meiotic male germ cells. *Epigenetics & Chromatin* 9, 1–18. <https://doi.org/10.1186/s13072-016-0099-8>
- Sin, H.-S., Kartashov, A.V., Hasegawa, K., Barski, A., Namekawa, S.H., 2015. Poised chromatin and bivalent domains facilitate the mitosis-to-meiosis transition in the male germline. *BMC Biol* 13, 53. <https://doi.org/10.1186/s12915-015-0159-8>
- Soumillon, M., Necsulea, A., Weier, M., Brawand, D., Zhang, X., Gu, H., Barthès, P., Kokkinaki, M., Nef, S., Gnirke, A., Dym, M., de Massy, B., Mikkelsen, T.S., Kaessmann, H., 2013. Cellular source and mechanisms of high transcriptome complexity in the mammalian testis. *Cell Rep* 3, 2179–2190. <https://doi.org/10.1016/j.celrep.2013.05.031>
- Vlaming, H., van Leeuwen, F., 2016. The upstreams and downstreams of H3K79 methylation by DOT1L. *Chromosoma* 125, 593–605. <https://doi.org/10.1007/s00412-015-0570-5>
- Wille, C.K., Sridharan, R., 2022. Connecting the DOTs on Cell Identity. *Front Cell Dev Biol* 10, 906713. <https://doi.org/10.3389/fcell.2022.906713>